

**BAYESIAN PHYSICAL-STATISTICAL RETRIEVAL OF SWE AND SNOW DEPTH FROM X**
**AND KU-BAND SAR - DEMONSTRATION USING AIRBORNE SNOWSAR IN SNOWEX'17**
Siddharth Singh[1], Michael Durand,[2] Edward Kim[3], Ana P. Barros[1]
[1]Department of Civil and Environmental Engineering, University of Illinois at Urbana-
Champaign, Urbana, Illinois, USA
[2]School of Earth Sciences, Ohio State University, Columbus, Ohio, USA
[3]NASA Goddard Space Flight Center, Greenbelt, Maryland, USA
*Correspondence to*: Ana P. Barros (barros@illinois.edu)
**Abstract**
A physical-statistical framework to estimate Snow Water Equivalent (SWE) and snow depth from
SAR measurements is presented and applied to four SnowSAR flight-line data sets collected
during the SnowEx'2017 field campaign in Grand Mesa, Colorado, USA. The physical (radar)
model is used to describe the relationship between snowpack conditions and volume backscatter.
The statistical model is a Bayesian inference model that seeks to estimate the joint probability
distribution of volume backscatter measurements, snow density and snow depth, and physical
model parameters. Prior distributions are derived from multilayer snow hydrology predictions
driven by downscaled numerical weather prediction (NWP) forecasts. To reduce noise to signal
ratio, SnowSAR measurements at 1 m resolution were upscaled by simple averaging to 30 and 90
m resolution. To reduce the number of physical parameters, the multilayer snowpack is
transformed for Bayesian inference into an equivalent single- or two-layer snowpack with the same
snow mass and volume backscatter. Successful retrievals, defined by absolute convergence
backscatter errors $\leq 1.2$ dB and local SnowSAR incidence angles between $30^{o}$ and $45^{o}$ for X- and
Ku-band VV-pol backscatter measurements, were achieved for 75% to 87% for all grassland
pixels with SWE up to 0.7m and snow depth up to 2 m. SWE retrievals compare well with snow
pit observations showing strong skill in deep snow with average absolute SWE residuals of 5-7%
(15-18%) for the two-layer (single-layer) retrieval algorithm. Furthermore, the spatial
distributions of snow depth retrievals vis-à-vis LIDAR estimates have Bhattacharya Coefficients
above 94% (90%) for grassland pixels at 30 m (90 m resolution), and values up to 76% in mixed
forest and grassland areas indicating that the retrievals closely capture snowpack spatial variability.
Because NWP forecasts are available everywhere, the proposed approach could be applied to SWE
and snow depth retrievals from a dedicated global snow mission.





## 1. Introduction

The seasonal snowpack plays a critical role in climate and weather variability due to its role in the surface energy budget on account of its high albedo, and in the surface water budget providing temporary storage of frozen precipitation in the cold season until it melts in the warm season and becomes available as runoff. The water stored in the snowpack is measured by the Snow Water Equivalent (SWE), the depth of liquid water per unit area that would be released if the snowpack were to melt completely. It is the product of the specific gravity of snow with respect to water ($\rho_{snow}/\rho_{w}$) and the depth of the snowpack (SD). To map SWE in the cold season is to map snow water resources. To map onset of melt and snow wetness is to map the timing and geography of snow water resources availability. Climate variability and change with increasing air temperature, shifts in atmospheric moisture convergence patterns, and increases in the frequency of extreme events is already causing significant changes in frequency and patterns and timing of seasonal snow accumulation and melt with severe implications for water and food security in addition to cascading economic and ecosystem impacts (Huang and Swain, 2022; Musselman et al., 2021; Sturm et al., 2010).

The need to capture snowpack heterogeneity and dynamics tied to weather, climate, landcover and landform variability remains a chief challenge to developing a snow observing system at the spatial and temporal scales required to answer water cycle science questions and for societal decision-making. The potential for systematic snowpack monitoring in remote regions has long been investigated, including the integration of remote sensing measurements and physical models (e.g. (Martinec et al. 1991; Mote et al. 2003; Bateni et al. 2015; Li et al. 2017; Kim et al. 2019; Cao and Barros, 2023). Assimilation of radiance or backscatter is most powerful with a time series of observations. Time-series observations are available presently from tower measurements, albeit at the point scale of the tower footprint. Airborne observations can be used for mapping but typically occur once or twice during a winter season and over limited areas. A dedicated satellite mission is necessary to acquire time-series of measurements globally.

Presently, advances in radar technology and retrieval algorithms (Tsang et al. 2022), and especially the demonstrated capabilities of NewSpace satellite missions (Villano et al. 2020) make high spatial resolution of Synthetic Aperture Radar (SAR; 10's m ) Earth observations from space feasible in contrast to the challenges faced in the past (Rott et al. 2012). During the SnowEx'17 field campaign (Kim et al. 2017), a comprehensive data set consisting of airborne dual-frequency SAR (X- and Ku-band Synthetic Aperture Radar) backscatter measurements using the SnowSAR instrument (Macedo et al. 2020), the Airborne Snow Observatory (ASO, Painter et al. 2018) and a plethora of high-quality ground-validation measurements of snowpack properties and ancillary data (Table 1) offer an unprecedented opportunity to investigate the full potential of SAR toward developing the next generation of retrieval algorithms.

Due to the highly nonlinear snow physics and the time-varying stratigraphy of snowpacks, radiance or backscatter measurements depend on the vertical structure of snowpack physical properties such as snow density, snow temperature, and snow grain size in addition to SWE and snow depth. Thus, SWE and snow depth retrieval is an underdetermined problem. Physical-statistical approaches enable physically-based constraints to relate measurements to geophysical states and parameters and more directly solving Bayes' law (Berliner, 2003; Kuhnert, 2014). In



this manuscript, we propose, implement, demonstrate, and evaluate a general physical-statistical
framework to retrieve SWE from SnowSAR measurements across a heterogeneous landscape
during SnowEx'17.

**2. Previous Work**
**2.1 Forward Simulation - From SWE to Backscatter**
The advantage of SAR technology is the high-spatial resolution of its measurements, which is
necessary to capture the spatial heterogeneity and temporal variability of snowpack physical
processes (e.g. Deems et al. 2016; Mendoza et al., 2020; Manickam and Barros, 2020) as
demonstrated in forward simulations. Cao and Barros (2020, 2022; hereafter CB20 and CB22)
demonstrated the utility of a coupled multi-layer snow hydrology (MSHM) coupled with a
radiative transfer model (RTM) forced by high-resolution operational numerical weather
prediction (NWP) model forecasts to capture the seasonal hysteresis behavior of the seasonal
snowpack at Grand Mesa and Senator Beck in Colorado against Sentinel-1 C-band measurements.
The MSHM is a physically driven snow hydrology model that simulates the evolution of snowpack
physical properties including detailed stratigraphy (Kang and Barros, 2012a-b). During snowfall
events, fresh snow is added to the top layer of the snowpack until a threshold accumulation is met,
and a new layer forms. The RTM used here is MEMLS3a (Microwave Emission Model of Layered
Snowpacks adapted to include backscattering by Proksch et al., 2015). MEMLS is a physically
driven radiative transfer model which takes snowpack characteristics as inputs and simulates its
microwave emission for a frequency band with four polarizations – HH, VV, HV and VH
(originally proposed by Wiesmann and Mätzler, 1999). To estimate total scattering, ground
backscatter $\sigma_{bkg}$ must be modeled as well, as described below. .
Figure 1 illustrates the various backscatter mechanisms contributing to total backscatter ($\sigma_{total}$) in
active microwave measurements represented in MEMLS3&a, the RTM: volume backscatter ($\sigma_{vol}$)
from the multiple interactions of the incoming radar signal within the snowpack, the backscatter
at the snowpack-air interface ($\sigma_{surf}$) and at the snowpack-ground interface including interactions
with submerged vegetation and litter ($\sigma_{bkg}$). In forested areas, additional backscatter mechanisms
are associated with the multiple bounce pathways among tree canopy, intercepted snow, tree
trunks, and snowpack. Depending on viewing geometry (flight path and incidence angle), $\sigma_{total}$
measurements from areas without trees in regions of mixed landcover can include significant
contribution from trees along the grassland-forest transitions.





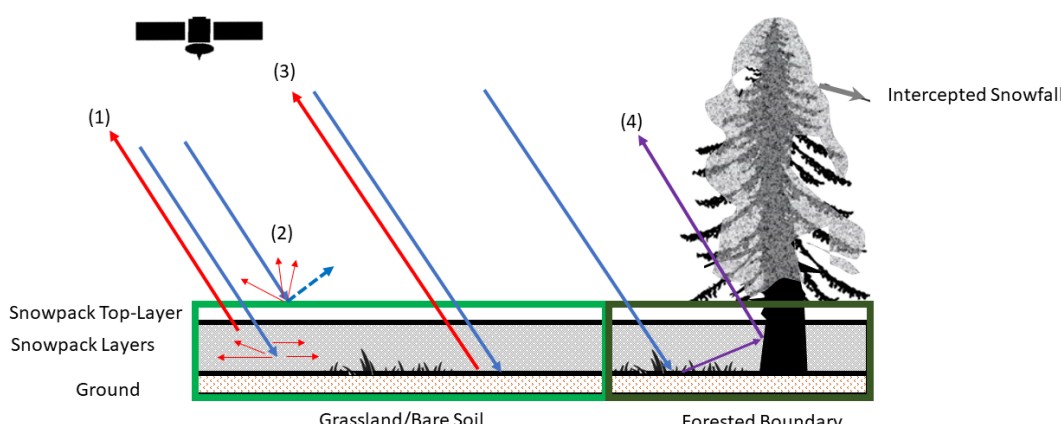


**Figure 1:** Scattering mechanisms for grassland submerged by snow and snowpack over bare soil or rock: (1) Volume Backscatter $\sigma_{vol}$; (2) surface backscatter $\sigma_{surf}$; (3) background backscatter at the snow-ground interface $\sigma_{bkg}$; (4) snowpack-ground-canopy-tree trunk interactions at forested boundaries. Red arrows (1), (2) and (3) are resolved in the retrieval applications demonstrated here.

CB22 used the coupled MSHM-MEMLS in forward mode to predict Sentinel-1 C-band volume backscatter $\sigma_{vol}$ without calibration or nudging of ground observations without bias and within ± 2.5 dB at 90 m resolution across terrain slopes in the [10°-52°] range for barren land and alpine grass and shrubs and in forested areas with snow-free canopy at the beginning of spring in the Senator Beck Basin in Colorado. They estimated $\sigma_{bkg}$ as the average of Sentinel-1 measurements for snow-free conditions. Cao and Barros (2023) modified MEMLS3&a to include double-bounce effects among snowpack and vegetation (MEMLS-V) and retrieved $\sigma_{bkg}$ from total backscatter $\sigma_{total}$ measurements in mixed landcover using simulated annealing. Their estimates are consistent with CB22, suggesting potential to simplify the inverse-problem of estimating snowpack physical properties from total backscatter measurements in mixed landcover and further simplfy the physical-statistical retrieval framework proposed here, although further evaluation is necessary.

**2.2 Physical-Statistical Retrieval**

For retrieval in a Bayesian framework, the probability of the retrieved geophysical variable *x* (the inferred posterior distribution) is conditional on the *a priori* knowledge of *x* (the prior distribution), indirect measurements D, and a physical model $M(\eta)$ (e.g., the snow radiative transfer algorithm in this case ) with physical parameters $\eta$ (including *x*) and statistical error parameters $\zeta$. The joint probability distribution of M, *y*, $\eta$, and $\zeta$ can be written as:

$$P(M, D, \eta, \zeta) = P(D|M, \eta, \zeta) \times P(M|\eta, \zeta) \times P(\eta, \zeta) \tag{1}$$

The first term to the right-hand side of Eq. (1) is the backscatter data model, the second term is the prior of the backscatter, and the third term is the prior of the snowpack physical parameters (including snow depth and snow density, etc) with statistical error parameters. Assuming the




measurements do not depend on the physical parameters, the model does not depend on the statistical error parameters, and that the physical parameters and the statistical parameters are independent, Eq. (1) can be revised to read

$$P(M, D, \eta, \zeta) = P(D|M, \eta) \times P(M|\eta) \times P(\eta) \times P(\zeta) \tag{2}$$

And finally in the context of specific measurements *y*

$$P(M, \eta, \zeta \,|\, y) = P(y|M, \eta) \times P(M|\eta) \times \frac{P(\eta) \times P(\zeta)}{P(y)} \tag{3}$$

The physical model M and P(*y*) are invariant and assuming that we have good understanding of the statistical errors, then Eq. (3) can be further simplified as follows

$$P(\eta|y) \propto P(y|\eta) \times P(\eta) \tag{4}$$

In the context of Bayesian inference the goal is to maximize P(η|y), the posterior probability of physical parameters conditional on measurements informed by the a priori parameter probabilities P(η). To maximize P(y|η), the posterior of the backscatter conditional on physical parameters η, implies minimizing the difference between measurements *y* with known error covariance matrix $\Sigma_y$ and model predictions *M(η)*. For a multivariate normal distribution, Durand and Liu (2012) proposed

$$P(y|\eta) = (2\pi)^{\left(-\frac{N}{2}\right)} \,|\, \Sigma_y \,|^{-\frac{1}{2}} \, exp\left[-\frac{1}{2}(y - M(\eta))^T \Sigma_y^{-1}(y - M(\eta))\right] \tag{5}$$

where N is the number of measurements at a given location and time (e.g. backscatter at different frequencies

Pan et al. (2023, hereafter P23) adapted a Bayesian retrieval algorithm previously developed to estimate SWE from passive microwave measurements (Pan et al. 2017, hereafter P17 ) to active microwave, hereafter referred to as Base-AM. The snow radiative transfer algorithm in Base-AM is MEMLS, and the semi-empirical Dobson model is used to estimate the soil dielectric constant as a function of soil moisture and soil texture (Dobson et al. 1985; Hallikainen et al. 1985). A Monte Carlo Markov Chain iterative algorithm (Metropolis et al. 1953) is used to sample from P(η|y) starting from initial values. Here, the realistic snowpack predictions from MSHM-MEMLS are used to define the prior distributions of parameters and constrain the Bayesian retrievals.

**3. Study Area and Data**

**3.1 Study Area and Ancillary Data**

The study region is Grand Mesa, Colorado, a plateau that is 2,000 m above adjacent low-lying areas and is surrounded by ridges up to 500m in elevation (as depicted in Fig. 2). Grand Mesa





has an alpine climate, experiencing snowfall throughout the year except during the months of July
and August. Landcover is heterogeneous with grasslands in the west and a mix of evergreen and
deciduous forest to the east. Numerous wetlands are widespread across the Mesa, especially in the
transition from grassland to forest. The land cover data were obtained from the National Land Data
Assimilation System (NLDAS) and the National American Land Change Monitoring System
(NALCMS), both at 30 m resolution. The datasets were upscaled to 90m using nearest neighbor
interpolation. NLDAS is used to determine landcover type in the snow hydrology model.
NALCMS is used to upscale the evaluation data. Hourly albedo is derived from NLDAS at 12.5
km resolution. A summary of all the datasets used in this study is available in Table 1.

**Table 1:** Summary list of datasets used in the study.

| Data | Source | Spatial Resolution | | Temporal Resolution | | Date Range | Relevant Link |
|---|---|---|---|---|---|---|---|
| | | Initial | Final | Initial | Final | | |
| Rainfall Temperature Air Pressure Incoming SW radiation Incoming Longwave radiation Wind speed Humidity | HRRR | 3 km | 30 m, 90 m | 1 hr | 30 min | 9/1/2016 - 2/25/2017 | https://rapidrefresh.noaa.gov/hrrr/ |
| Albedo | NLDAS | 12.5 km | 30 m | 1 hr | 30 min | 9/1/2016- 2/25/2017 | https://ldas.gsfc.nasa.gov/ |
| Backscatter | SnowSAR – SnowEx'17 | 1 m | 30 m, 90 m | - | - | 2/21/2017 | https://nsidc.org/data/snex17_snowsar/versions/1 |
| Landcover | NLCD, NALCMS | 30 m | 30 m, 90 m | - | - | - | https://www.usgs.gov/centers/eros/science/national-land-cover-database http://www.cec.org/north-american-land-change-monitoring-system/ |
| Snow Depth | LIDAR – SnowEx'17 | 3 m | 30 m, 90 m | - | - | 2/25/2017 | https://nsidc.org/data/snex17_snowpits/versions/1 |
| SWE | Snowpit – SnowEx'17 | - | - | - | - | 2/20/2017- 2/24/2017 | https://nsidc.org/data/aso_3m_sd/versions/1 |

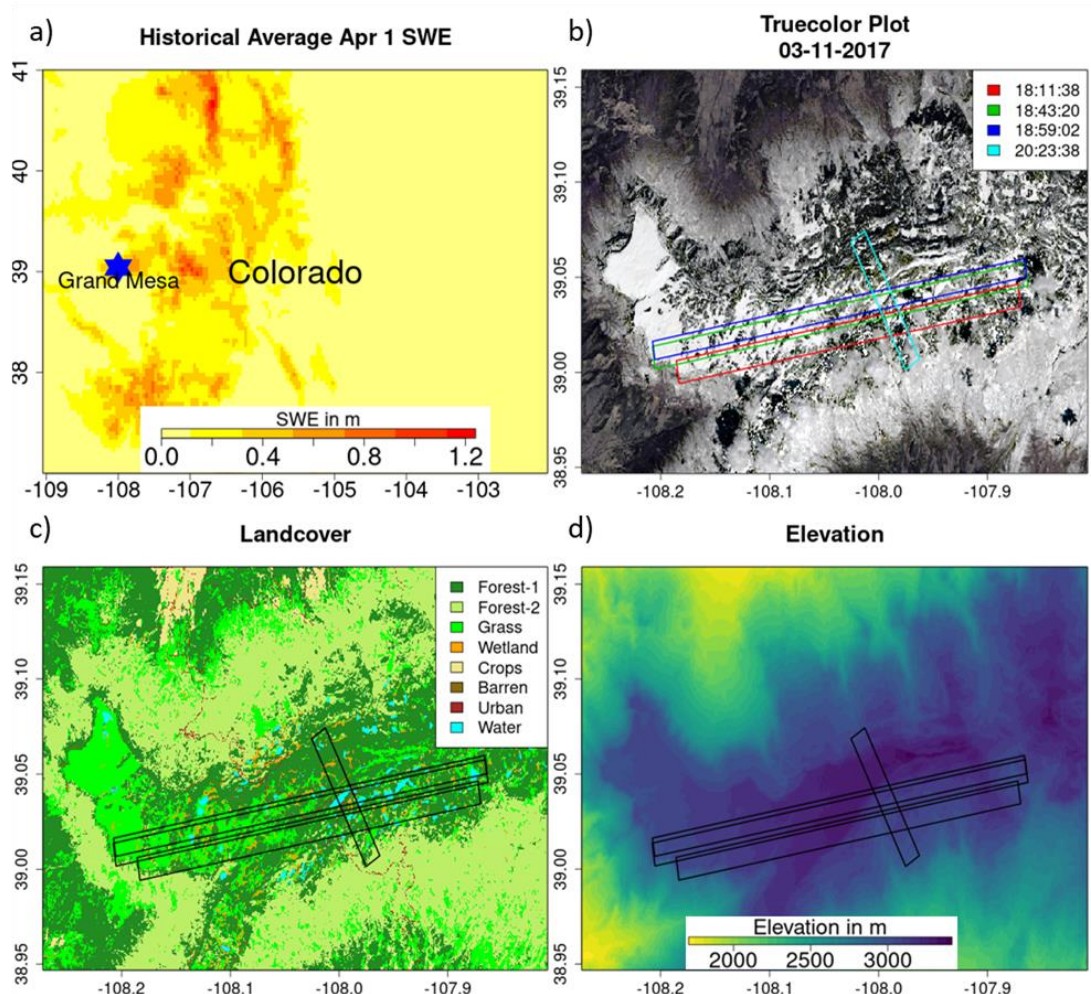

**Figure 2:** Study area in Grand Mesa, Colorado. a) Location of Grand Mesa in Colorado, with historical Apr 1 SWE average as base map. b) Paths of 4 SnowSAR SnowEx'17 flights on 21 Feb 2017, with true color image obtained from Landsat on 03/11/2017 as the base map. c) Land cover of the study region. Forest-1 are needle leaf forests; Forest-2 are broadleaf forests. d) Digital elevation map of the study region.

## 3.2 Atmospheric Forcing

Numerical Weather Prediction (NWP) outputs are used as the atmospheric forcing for the snow hydrology model and to set up boundary conditions. Previously, CB20 and CB22 relied on HRRR (High-Resolution Rapid Refresh) hourly forecasts at 3 km and downscaled it to 90 m in Grand Mesa. Here, the same data set was independently downscaled to 30 m as well. The HRRR dataset is produced by National Ocean and Atmospheric Agency (NOAA) by hourly assimilation of observations at 13 km resolution (Benjamin et al., 2016; Table 1). Hourly atmospheric forcing was linearly interpolated to 30 min temporal resolution used in the snow hydrology model.



196

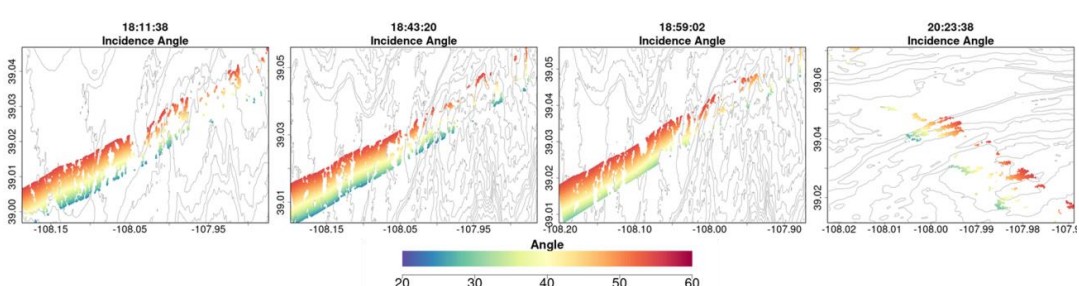

197

**Figure 3:** Maps of incidence angles along SnowSAR flight paths on February 21, 2017 during SnowEx'17.

199

## 3.4 SnowSAR Backscatter

During SnowEx'17, airborne microwave backscatter measurements were made in Grand Mesa on 21 Feb 2017 at 1 m resolution (Table 1). The SnowSAR instrument is a dual frequency (X and Ku Band) radar. A total of six flightlines were completed, two short ones on sloped densely forested terrain and four long lines on the plateau. Here, only the four flightlines on the plateau are used for analysis (Fig. 2 and Fig. 3). The flights are between 18:00 and 21:00 GMT (noon – 3PM  MST). SnowSAR data quality control measures included filtering based on aircraft attitude (there were line segments with turbulence), beam incidence angle/antenna pattern, and signal-to-noise-ratio of the backscatter coefficients. Processing of the original airborne SAR measurements and quality control indicate that only the co-pol X-band HH- and VV-pol as well as Ku-band VV-pol measurements are adequate for retrieval. Geolocation was verified against corner reflector targets and geographic features and found to be very robust. The SnowSAR data were upscaled to 30 m and 90 m resolution by simple averaging of all SnowSAR measurements within each pixel.

213

## 3.5 Validation Data

*LIDAR Snow Depth* – The ASO LIDAR measurements of snow depth at 3m resolution across Grand Mesa  are available for SnowEx'17 on February 25,  thus 4 days after the SnowSAR flights (Painter et al., 2018; Table 1). There were no significant snow storms or strong winds in that period, except for about 3mm of rainfall for less than 1 hour on February 24 th.  These data are used to examine the distribution of retrieved snow depths, that is indicative of the spatial heterogeneity of the snowpack, and the relative absolute  differences between LIDAR measurements and retrieval of snow depth,  that are indicative of local retrieval errors.  The LIDAR data were upscaled to 30 m and 90 m using simple averaging  (e.g., Fig.4a). There can be large snow depth underestimation errors associated with upscaled LIDAR retrievals along the edges of forests where the snow depth is underestimated consistent with previous work (e.g. Deems et al. 2013; Jacobs et al. 202).  Given the expect measurement uncertainty on the order of 10-20 cm in Grand Mesa, which is amplified by microtopography, LIDAR pixels with snow depth shallower than 20 cm are not considered for evaluation.





*Snowpit SWE* - Multiple snowpits were excavated during the SnowEx'17 field campaign across
Grand Mesa (Table 1). Due to the small number of snow pit measurements along the SnowSAR
flightlines on 21 February,  snowpit measurements on 20-24 of February were considered for
evaluation assuming that in the absence of  snowstorms or other weather events the snow pack
does not change significantly during the 4-day period. Differences are expected at local places but
the overall spatial trends should be maintained such as the west-east gradient in snow depth.  The
values of snowpit SWE are estimated using an average of the snow density measurements at
different depths applied to the entire snow depth. Only pits in  non-forested areas were selected
for evaluation (Fig. 4b).

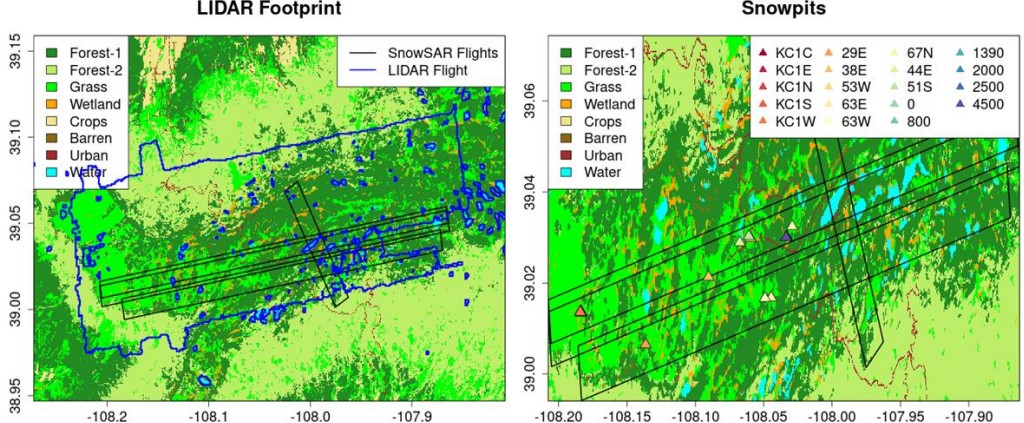

**Figure 4:**   a) Flight footprint of the LIDAR instrument used to measure the snow depth during SnowEx'17. b) Location of snow
pits used to measure SWE 20-24 Feb 2017.  The legend identifies SnowEx'17 Pit IDs.

**4. Methods**
**4.1 Retrieval Algorithm**
The retrieval methodology builds on existing and well evaluated snow hydrology, radiative
transfer,  and physical-statistical models (CB20,CB22, P17, P23) previously reviewed in Section
2. Figure 5  illustrates the retrieval workflow consisting of four main steps: **(1a)** Snow hydrology
simulation using MSHM to produce a layered snowpack; **(1b)** Volume backscatter $\sigma_{vol}$ simulation
using MEMLS  and estimation of  background backscatter  $\sigma_{bkg}$ by substration from SnowSAR
$\sigma_{total}$ measurements;  **(2)** Determination of  snowpack parameter prior distributions for retrieval:
averaged snowpack physical property distributions for a 1 or 2 layer equivalent snowpack (1|2)
with the same mass and total backscatter $\sigma_{total}$ ; **(3)** Determination of ground priors for retrieval:
Bayesian estimation of  ground parameters that govern the $\sigma_{bkg}$ using MEMLS for a very thin (1
mm SD) film of snow on the ground; and **(4)** Retrieval: Bayesian optimization of simulated $\sigma_{total}$





to derive posterior distributions of SD and $\rho_{snow}$ for the 1|2 equivalent snowpack, and subsequent
calculation of SWE.

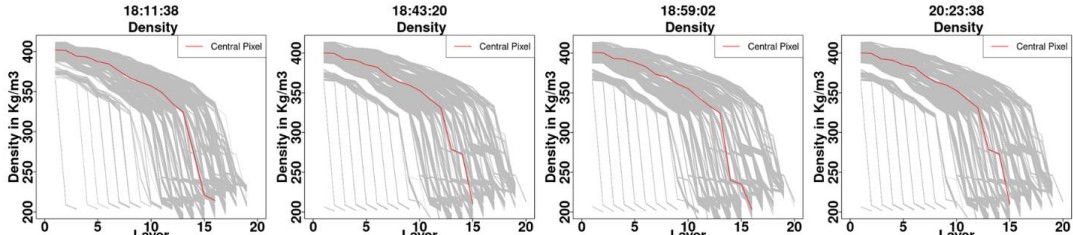


**Figure 5** - Density profiles obtained from MSHM for the 4 SnowSAR flight paths. The density profile of the central pixel for each
of the flights is marked in red. Note the significant difference between the top 2-3 layers and the deeper snowpack supporting the
two-layer snowpack concept.

## 4.1.1 Layered Snowpack Simulations and Prior Distributions (1a,2)

MSHM was run for a full-year starting from snow free conditions on September $1^{st}$ 2016 using
downscaled HRRR data as atmospheric forcing (Section 1.2) and a timestep of 30 mins. On the
day of the SnowSAR flights, the snowpack physical properties predicted at times corresponding
to each of the four flights are used to derive the 1|2 Layer equivalent snowpack properties used in
the retrieval. Volume backscatter was estimated using the cross polarization fraction Q=0.2. The
prior distributions for Base-AM are shown in Table 3.
In realistic layered snowpacks, stratigraphy (i.e., vertical heterogeneity) is a dominant feature of
the density, temperature, microstructure, and dielectric properties (e.g., emissivity and
reflectivity). The vertical structure of snow microphysics in MSHM is described using a
parameterization of snow correlation length consistent with MEMLS formulation. Depending on
the number of layers, this poses an overdetermined problem as the number of measurements is
equal to the number of frequencies and the number of polarizations available (typically two or
three). For example, there are only four observations for a dual-frequency measurement with dual
polarization. In contrast, the set of independent parameters per layer includes snow density, layer
thickness, liquid water content, snow grain size or correlation length, temperature, reflectivity, and
transmissivity. To reduce the number of independent parameters that need to be estimated, the
multilayer snowpack is transformed into an equivalent single- or two-layered snowpack with the
same SWE, snow depth (SD) and total backscatter $\sigma_{total}$.








**Table 2:** Input and output parameters from the three models in the SWE physical-statistical retrieval framework.

| Model | Input | Output | Reference |
|-------|-------|--------|-----------|
| MSHM | Rainfall<br>Temperature<br>Air Pressure<br>Incoming shortwave radiation<br>Incoming longwave radiation<br>Wind speed<br>Humidity<br>Albedo | Snow Temperature Profile<br>Soil Temperature Profile<br>Snow Density Profile<br>Snow Depth Layering Profile<br>Liquid Water Content Profile<br>Snow Correlation Length Profile | Cao and Barros (2020) |
| MEMLS | Snow Temperature Profile<br>Soil Temperature Profile<br>Snow Density Profile<br>Snow Depth Layering Profile<br>Snow Correlation Length Profile<br>Cross polarization fraction<br>Ground rms height | Diffused Reflectivity Profile<br>Specular Reflectivity Profile<br>Total Backscatter Coefficient | Proksch et al. (2015) |
| Base AM | Equivalent Snow Temperature Prior<br>Equivalent Soil Temperature Prior<br>Equivalent Snow Density Prior<br>Equivalent Snow Depth Prior<br>Correlation Length<br>Cross polarization fraction<br>Ground rms height<br>Total Backscatter Coefficient Prior | Optimized – Snow Layer Depth<br>Snow Density | Pan et al., 2023 |

**Table 3:** Base-AM model input standard deviation and range for the lognormal parameters prepared using MSHM multilayer snowpack parameters. Alphanumerical subscript in 2-layer snowpack retrievals denotes layer number: 1- bottom layer; 2- top layer; avg- the average of all MSHM multilayer parameter values in the corresponding single or 2-layer snowpack. DZ is the MSHM snow depth.

| Snow Parameters | 1 Layer Snowpack | | | 2 Layer Snowpack | | | |
|-----------------|------------------|----|----|------------------|----|----|----|
| | Standard Deviation, σ | Range | | Standard Deviation, σ | | Range for each layer | |
| | | Min | Max | Bottom | Top | Min | Max |
| Snow Temp., Ts [$^{\circ}$C] | $0.3\times Ts_{avg}$ | $1.3\times Ts_{min}$ | $0.7\times Ts_{max}$ | $0.3\times Ts_{1,avg}$ | $0.3\times Ts_{2,avg}$ | $1.3\times Ts_{min}$ | $0.7\times Ts_{max}$ |
| Snow Density, ρ [$Kg/m^3$] | $0.3\times\rho_{avg}$ | $0.8\times\rho_{min}$ | $1.2\times\rho_{max}$ | $0.3\times\rho_{1,avg}$ | $0.3\times\rho_{2,avg}$ | $0.8\times\rho_{min}$ | $1.2\times\rho_{max}$ |
| Snow Depth, DZ [m] | $0.3\times DZ$ | $0.5\times DZ$ | $1.5\times DZ$ | $0.1\times DZ_1$ | $0.2\times DZ_2$ | $0.2\times DZ$ | $0.9\times DZ$ |
| Correlation Length, D | $0.3\times D_{avg}$ | $D_{min}$ | $D_{max}$ | $0.2\times D_{1,avg}$ | $0.2\times D_{2,avg}$ | $D_{min}$ | $D_{max}$ |
| Soil Temp., Tsoil [$^{\circ}$C], | 0.3 | 1.3 | | 0.3 | | 1.3 | |

*Single-layer Snowpack -* The total snow depth and the averages of multilayered snowpack parameters are specified as the mean of the prior distribution for retrieval. Table 3 shows the range and standard deviation of the parameters.



*Two-layer Snowpack* – The average values of the snowpack physical properties for each of two
layers are derived from the multilayer snowpack simulated by MSHM as for the single-layer case.
The key requirement is to determine the depth of each one of the layers that best captures the
snowpack vertical structure. Figure 5 shows examples of MSHM snow density profiles for each
of the four SnowSAR flights. Note the large changes with depth of the snow density profiles. The
shape of the profiles reflects the interplay between thermodynamic processes that change snow
microstructure and dominate in the upper snowpack and mechanical consolidation processes that
are dominant in the mid and lower layers. The snow depth point corresponding to the maximum
change in snow density between adjacent layers in the multilayer snowpack is used here to divide
the snowpack in two layers. Subsequently, the average density, snow temperature, and correlation
length of the MSHM multilayer snowpack is calculated for the corresponding depths of the two-
layer equivalent snowpack (Table 3).

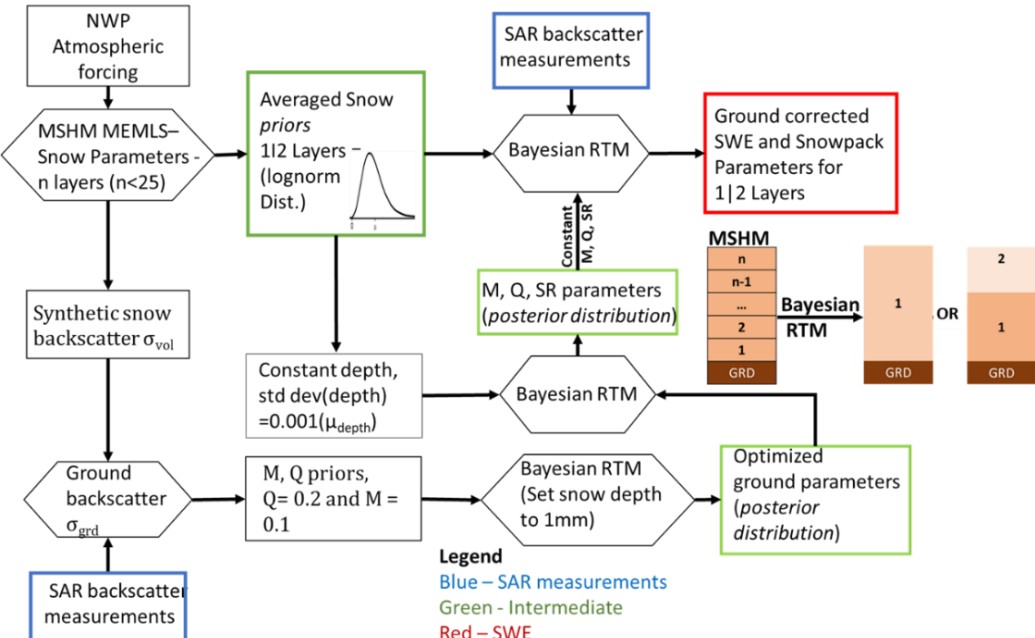

**Figure 6:** Workflow of the SWE Physical-Statistical retrieval. NWP atmospheric forcings are used to set up MSHM to determine
priors for the Bayesian radiative transfer model (Base-AM) and synthetic backscatter for ground parameters. SnowSAR backscatter
measurements are assimilated to determine the posterior distribution of the snowpack parameters.

## 4.1.2 Determination of Ground and Snowpack Microstructure Parameters (1b, 3)

A first estimate of the $\sigma_{bkg}$ is obtained by subtracting $\sigma_{vol}$ from SnowSAR X-band HH- pol $\sigma_{total}$
measurements. In Base-AM, $\sigma_{bkg}$ depends on the effective effective soil moisture and soil surface
roughness. To optimize these parameters, $\sigma_{bkg}$ is used as an "observed" value. To simulate snow-




free conditions the snow depth is constrained to a maximum value of 1 mm. The cross polarization
fraction Q initially specified as Q=0.2 is optimized first and separately from other ground
parameters in the third step of the retrieval algorithm (Fig. 6). Finally, the posterior distributions
of the ground parameters are used along with the 1|2 layer prior distributions and the SnowSAR
measurements to estimate the posterior distributions of snow depth and snow density using the
Base-AM framework (Fig. 6) and both X- and Ku-band VVpol. SWE is subsequently derived from
snow depth and snow density.

**4.2 Retrieval Evaluation**
The local mean of the posterior distribution for each parameter is hereafter referred to as the
retrieval result for each pixel. The retrievals are evaluated against LIDAR snow depth including
spatial patterns and gradients, and overall statistical structure using histograms. SWE retrievals
derived from the posterior distributions of snow density and snow depth are evaluated against SWE
measurements at snowpits (Section 3). Original LIDAR measurements were reprojected and
coregistered with the SnowSAR retrievals. A comparative analysis retrievals was conducted to
examine the dependence of retrievals on incidence angle for different levels and the subgrid scale
variability quantified as the standard deviation of original LIDAR measurements within the
upscaled pixel. The amplitude error metrics are the mean, standard deviation, and mean absolute
relative error (MARE):
$$MARE = \frac{\sum_{i=1}^{n} |1 - R_i/O_i|}{N} \tag{6}$$

where O are observations and R are retrievals. The Bhattacharya coefficient (BC) is used to
compare the spatial distributions of snow depth and backscatter. BC measures the similarity
between two probability distributions $p_1$ and $p_2$ as follows ( Bhattacharya, 1943)
$$BC = \sum_{i=1}^{N} \sqrt{p_1(i)p_2(i)} \tag{7}$$

Finally, among the 39 snowpits available for evaluation on February 21, only 15 pits in open areas
(i.e. grasslands) were retained for evaluation and snow pits without SnowSAR measurements
within a radius of 100 m were discarded.

**5. Results and Discussion**
**5.1. Successful Retrievals**
SnowSAR measurements are strongly affected by aircraft operations, viewing geometry that varies
systematically along the flight path resulting in amplitude artifacts amplified by landform and
landcover heterogeneity. Even after separating homogeneous grassland pixels, there is
contamination from multiple bounce artifacts at grassland-forest transitions and adjacent wetlands
that cannot be resolved at 30 or 60 m resolution. Other errors embedded in the retrieval are





associated with downscaling of HRRR forcings that produce biased snow priors, snow hydrology
model assumptions, and errors tied to the background backscatter estimation. Combined these
errors can lead toweak convergence of the Bayesian optimization algorithm resulting in large
backscatter residuals. To account for these errors, SnowSAR pixels for which the relative residual
backscatter (RRB) between Base-AM simulated $\sigma_{total}$ and SnowSAR measurements was greater
than 30% were identified as unsuccessful. In an operational context, these pixels would be flagged
and identified as failed or highly uncertain retrievals. The successful retrieval fraction after
restricting the range of incidence angles and imposing the RRB < 30% criterion is summarized in
Table 4 for the four flights, and 1|2 layer snowpack retrievals at 30 and 90 m resolution. Except
for the later flight path over the predominantly forested areas in the eastern sector of Grand Mesa
(Fig.1), the fraction of successful retrievals by restrictingthe incidence angle and RRB varies
between 75 and 87% across the four SnowSAR flights with a maximum absolute bias of 1.2 dB.

**Table 4**: Spatial bias between SnowSAR backscatter and converged backscatter from Base-AM for successful retrievals for
grassland pixels at 30 and 90 m spatial resolution over each flight. Successful retrievals are for pixels with local incidence angles
in the 30º-45º range and relative residual backscatter (RRB) of less than 30% for each of the four flights. Shaded columns are for
retrievals at 90 m resolution.

| Flight Time | Successful Retrieval Fraction | | | | Bias (Observed - Converged) [dB] | | | | | | | |
|---|---|---|---|---|---|---|---|---|---|---|---|---|
| | 1 Layer | | 2 Layer | | 1 Layer | | | | 2 Layer | | | |
| | | | | | 30 m | | 90 m | | 30 m | | 90 m | |
| | 30 m | 90 m | 30 m | 90 m | X | Ku | X | Ku | X | Ku | X | Ku |
| 18:11:38 | 0.86 | 0.87 | 0.85 | 0.86 | 0.92 | -0.45 | 0.96 | -0.48 | 0.94 | -0.46 | 0.97 | -0.50 |
| 18:43:20 | 0.75 | 0.75 | 0.75 | 0.75 | 1.08 | -0.54 | 0.98 | -0.36 | 1.07 | -0.46 | 0.98 | -0.37 |
| 18:59:02 | 0.78 | 0.81 | 0.81 | 0.81 | 1.20 | -0.78 | 1.21 | -0.79 | 1.15 | -0.73 | 1.22 | -0.83 |
| 20:23:38 | 0.66 | 0.69 | 0.57 | 0.69 | 0.51 | -0.58 | 0.70 | -0.43 | 0.62 | -0.85 | 0.72 | -0.45 |



## 5.2. Retrieval Skill

Figure 7 compares LIDAR snow depth (Fig. 7a) against colocated SnowSAR retrievals at 30 m
for the SNOWSAR flight at 18:11:38 GMT(GMT=MST+6). The SnowSAR retrievals for high
incidence angles underestimate the LIDAR snow depth (orange and red points). Lemmetyinen et
al. (2022) suggested a nominal incidence angle of 35º-45º for retrievals ensuring proper focusing
and calibration of SnowSAR swaths. CB22 showed good skill in forward backscatter simulations
for incidence angles as low as 30º. Overall the retrievals here also show very good performance
for incidence angles between 30º-45º. Note however the large residuals for SnowSAR retrievals
with high incidence angles (red and orange points in Fig. 7b) corresponding to LIDAR pixels with
shallow snow depth (below the 1:1 line) and large subgrid-scale variability (orange and red points,
Fig. 7c). Analysis for all flights at both 30 and 90 m resolution can be found in Appendix A (
please see Figs. A1 and A2 similar to Fig. 7b; and Figs. A3 and A4 similar to Fig. 7c). Figures
7d, 7e, and 7f show the landcover, spatial distribution of subgrid standard deviation and absolute



residual (Retrieved – LIDAR) snow depth for the same flight.  Along the edges of forest, the
standard deviation in the upscaled pixels is large due to high heterogeneity that cannot be resolved
by the the LIDAR fusion algorithm for snow depth retrieval (Painter et al. 2016). The red box
identifies an area with complex grassland-forest  boundaries (Fig. 7d) and high subgrid scale
variability (Fig. 7e) resulting in poor LIDAR estimates. The edge of wetlands also has
comparatively higher residuals than completely homogeneous grasslands.  This corresponds to the
LIDAR pixels with standard deviation of more than 0.3 m (yellow, orange and red in Fig. 7c).
Therefore, only LIDAR pixels with subgrid-scale standard deviations ≤  0.3m  are used for
assessment of retrievals.

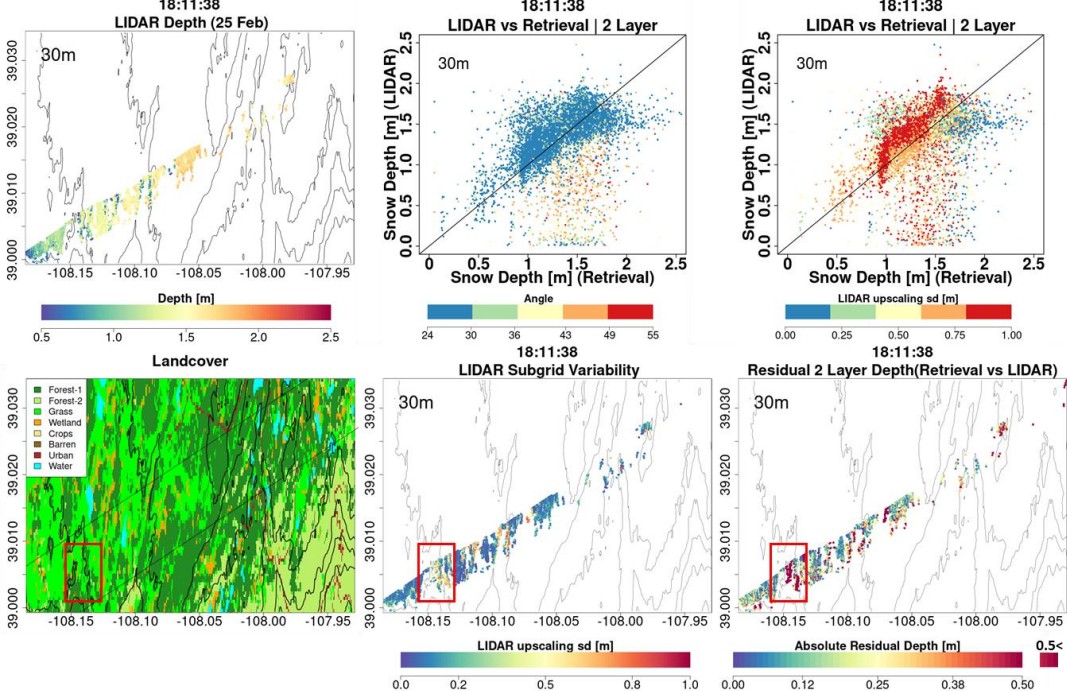


**Figure 7:** a) Snow depth measurements using airborne LIDAR on 2/25/17, 4 days  after the SnowSAR flights. b) Comparison
between LIDAR snow depth and the retrieved snow depth for 2-layer snowpack for the SnowSAR on 2/21/17 at 18:11:38 GMT.
The pixels are color-coded according to the incidence angle for the SnowSAR observations. c) The same comparison is shown;
however, the pixels are color-coded  according to the subgrid-scale variability of LIDAR snow depth within the corresponding 30
m pixel. Pixels on the edge of forests and grassland have higher standard deviations. d) Landcover distribution along the flight
path. e) Spatial distribution of subgrid-scale variability of upscaled LIDAR snow depth at 30m corresponding to part c). The edges
of forests have higher subgrid-scale variability due to errors in the LIDAR snow depth retrievals at high resolution. f) Absolute
residual between retrieved and LIDAR snow depth. Residuals equal to 0.5 m and above are grouped in the same category. The red
box in the parts d), e), and f) delineates an area with large absolute residuals. The areas on the edge of the forests have large subgrid-
scale variability in the LIDAR retrievals contributing and there are vegetation-snowpack backscatter interactions that are not
accounted for in the  retrievals. Additionally, areas surrounding the wetlands have comparatively higher residuals than the
homogenous grasslands.





Figures 8 and 9 show heatmaps to compare successful retrievals and observed X-band and Ku-
band VV-pol total backscatter at 30 m resolution.  There is good agreement between the two values
for both the bands specially for -15 dB to -10 dB  without significant differences between single
and two-layer snowpack retrievals. There is a constant positive bias in case of X-band simulations
compared to observations, whereas Ku-band has a constant negative bias as quantified in Table 4.
Overall, the retrievals at 90 m resolution show better agreement than those at 30 m resolution due
to averaging ( Figs. A5 and A6).

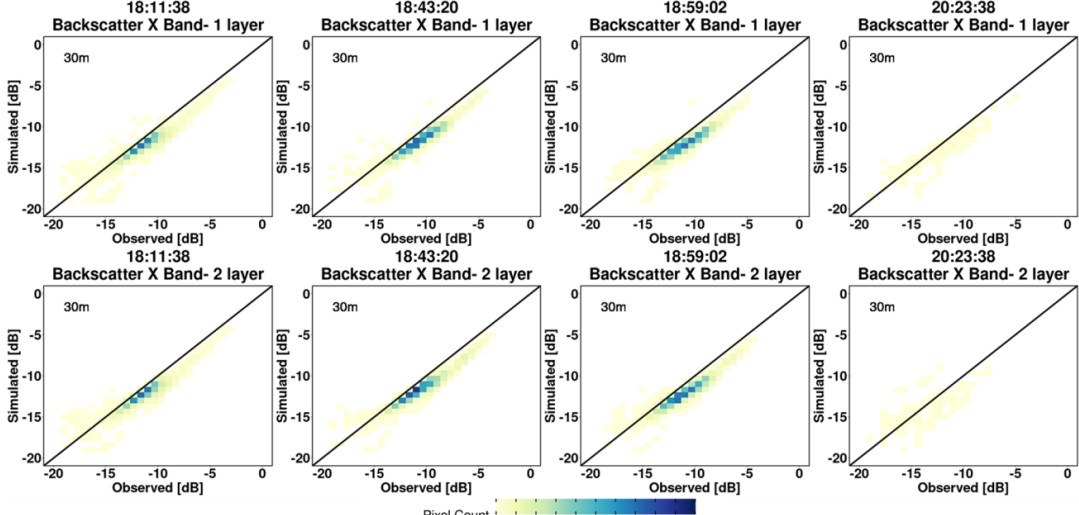


**Figure 8:** Heatmaps of SnowSAR (observed) backscatter (X-band)  versus  converged (simulated) backscatter at 30 m resolution:
1-layer snowpack (top row); 2-layer snowpack (bottom row).  Successful retrievals are for pixels with local incidence angles in the
30º- 45º range and relative residual backscatter (RRB) of less than 30% for each of the four flights (see Table 4).



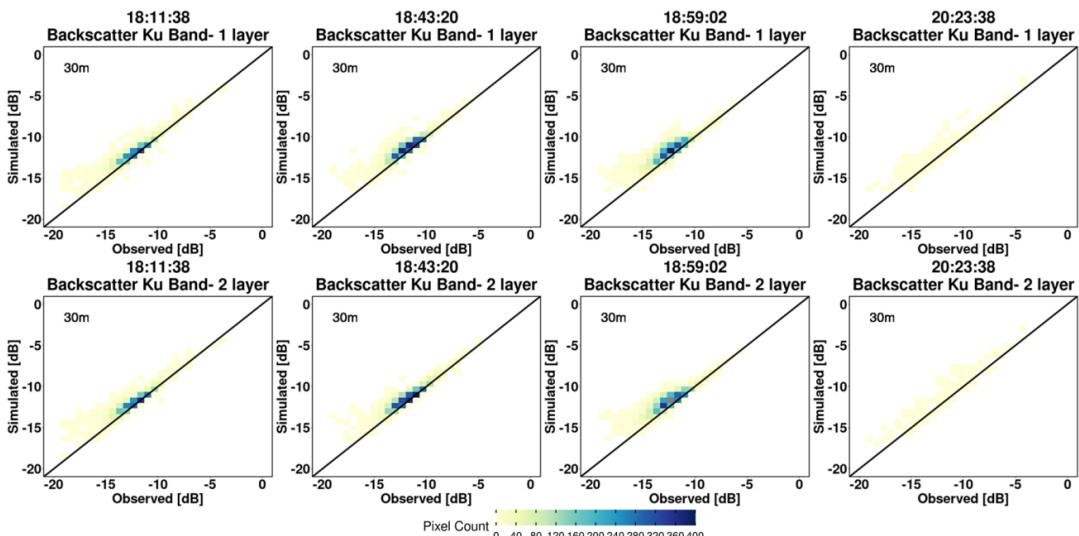

**Figure 9:** Heatmaps of SnowSAR (observed) backscatter (Ku-band) versus converged backscatter (simulated) for successful retrievals at 30 m resolution: 1-layer snowpack (top row) and 2-layer snowpack(bottom row). Successful retrievals are for pixels with local incidence angles in the 30º- 45º range and relative residual backscatter (RRB) of less than 30% for each of the four flights (see Table 4).

Maps of successful SWE retrievals for the four SnowSAR flight paths are shown in Fig. 10 and Fig. A7 at 30 m and 90 m resolution, respectively. The retrievals capture well the west-east gradient in SWE, and show realistic spatial variability across Grand Mesa. The very low SWE and shallower snow depths at the easternmost boundary of the flightlines are underestimates introduced by upscaling of the SNOWSAR backscatter values where there are significant changes in topography at the edge of the Plateau (see Fig.2).



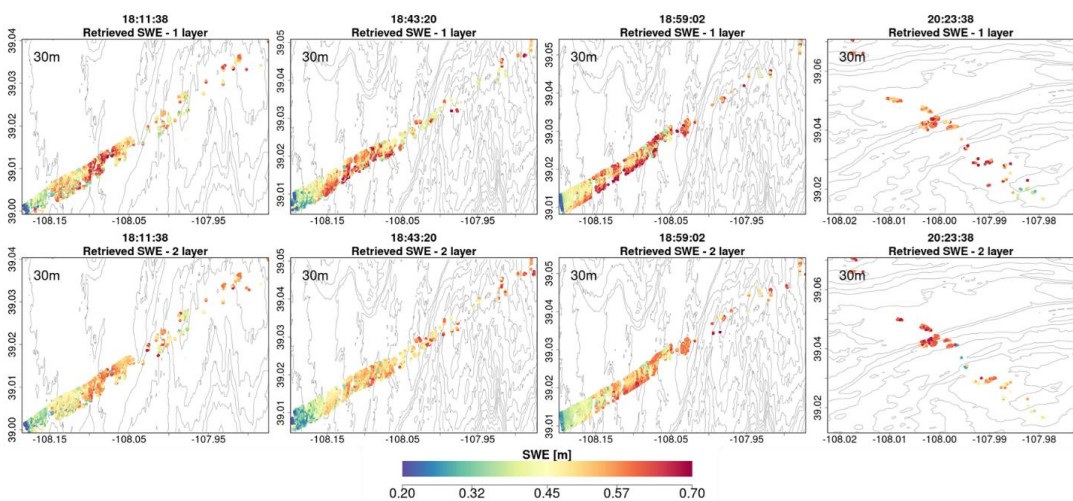

**Figure 10**: Spatial distribution of successful SWE retrievals for 1-layer and 2-layer snowpacks in grassland pixels at 30 m resolution. Successful retrievals are for pixels with local incidence angles in the 30°- 45° range and relative residual backscatter (RRB) of less than 30% for each of the four flights (see Table 4).

Heatmaps of total snow depth priors (MSHM predicted snow depth) against LIDAR snow depth are shown in Fig. 11 and Figs. A8 at 30 m and 90 m resolution and can be contrasted with heatmaps of total snow depth posteriors) against LIDAR snow depth in Figs. 12 and 13 using both single and two-layer retrievals. Note the narrow range of the prior snow depths concentrated around 1.5 m and the positive bias relative to LIDAR. The posteriors show much wider range of variability and deeper snow consistent with the LIDAR data for both single and two-layer retrievals, albeit with better agreement for the latter with high counts overlaying the 1:1 line at both spatial resolutions. This behavior is further confirmed by examining the snow depth histograms in Figs. A9 and A10 at 30 m and 90m resolution. The retrievals capture well the range of the LIDAR snow depths for all flights, and there is a substantial improvement in the shape of the distributions as revealed by the heatmaps.



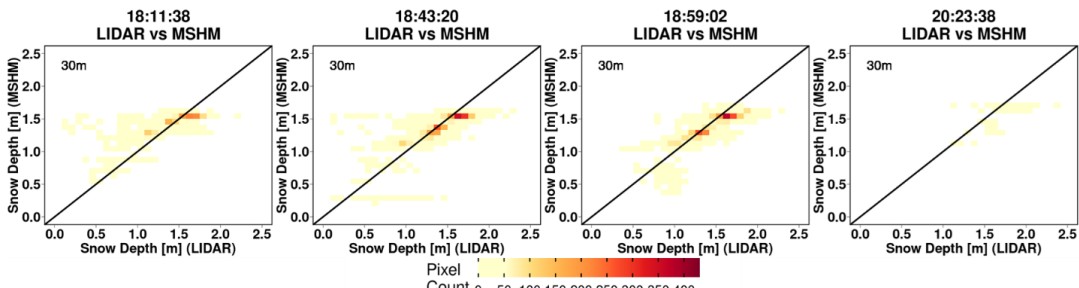


**Figure 11:** Heatmap of LIDAR and MSHM predicted snow depth priors at 30 m resolution using overlapping pixels from the MSHM and LIDAR. Only pixels with incidence angle between 30° -45°, and moderate sub-grid scale variability of LIDAR snow depth (< 0.3).


Quantitative assessment metrics are presented in Tables 5 and 6 for the comparison between
various snow depth datasets at 30 and 90 m resolutions, respectively. The snow depth MARE is
higher for the retrievals compared to the priors (MSHM) due to the fact that MARE is an effective
metric capturing distance from the mean. CB20 showed that the MSHM simulated average snow
mass accumulation at the Grand Mesa scale is within 10% of observations at a monthly time-scale
in February 2017. The BC coefficients show significant agreement between in the shapes of the
distributions at 0.95 or above at 30m resolution using the two-layer retrievals for the west-east
flights, and 0.76 for the fourth flight over the forest area at 20:23:38 GMT. There is significant
improvement in the statistical similarity of the snow depth retrievals vis-à-vis the LIDAR data for
all cases, and more so for the fourth flight. In all cases 30 m retrievals and two-layer retrievals
performed better than 90 m and single-layer retrievals for snow depth. This is explained in part
by landcover classification errors that are smaller at 30 m. Figure A11 shows that the number of
pixels where retrievals produce large mean absolute residuals is very small and characterize by
low confidence in the LIDAR estimates.



**Table 5:** Summary of statistics and  error metrics of the 3 snow depth (SD) data sets at 30 m resolution: LIDAR measurements,
MSHM predictions, and successful SnowSAR retrievals for grassland pixels and subgrid-scale standard deviation (σ ) of less than
0.3 m for the upscaled LIDAR pixel. MARE – Mean Absolute Relative Error (Eq. 6); BC – Bhattacharya Coefficient (Eq. 7). Here
mean and standard deviation refer to the spatial distribution, unlike the prior mean and standard deviation used in Base-AM (Table
3). Successful retrievals are for pixels with local incidence angles in the 30º- 45º range and relative residual backscatter (RRB) of
less than 30% for each of the four flights (see Table 4).

| Flight (GMT) | N Layer | Spatial SD μ [m] | | | Spatial SD σ [m] | | | MARE SD | | BC SD | |
|---|---|---|---|---|---|---|---|---|---|---|---|
| | | Retrieved | MSHM | LIDAR | Retrieved | MSHM | LIDAR | Retrieved-LIDAR | MSHM-LIDAR | Retrieved-LIDAR | MSHM-LIDAR |
| 18:11:38 | 1 | 1.39 | 1.42 | 1.42 | 0.32 | 0.15 | 0.28 | 0.19 | 0.11 | 0.94 | 0.67 |
| 18:43:20 | | 1.41 | 1.38 | 1.42 | 0.32 | 0.21 | 0.27 | 0.18 | 0.11 | 0.96 | 0.75 |
| 18:59:02 | | 1.49 | 1.38 | 1.44 | 0.33 | 0.20 | 0.27 | 0.18 | 0.09 | 0.94 | 0.76 |
| 20:23:38 | | 1.66 | 1.58 | 1.77 | 0.36 | 0.16 | 0.22 | 0.21 | 0.13 | 0.71 | 0.25 |
| 18:11:38 | 2 | 1.38 | 1.41 | 1.40 | 0.30 | 0.17 | 0.29 | 0.14 | 0.12 | 0.98 | 0.67 |
| 18:43:20 | | 1.35 | 1.38 | 1.42 | 0.31 | 0.20 | 0.28 | 0.14 | 0.11 | 0.97 | 0.75 |
| 18:59:02 | | 1.40 | 1.38 | 1.44 | 0.31 | 0.20 | 0.27 | 0.12 | 0.09 | 0.95 | 0.75 |
| 20:23:38 | | 1.89 | 1.61 | 1.80 | 0.39 | 0.14 | 0.24 | 0.17 | 0.12 | 0.76 | 0.23 |


**Table 6** - Summary of statistics and error metrics of the 3 snow depth (SD)  data sets at 90 m resolution: LIDAR measurements,
MSHM predictions, and successful SnowSAR retrievals for grassland pixels and subgrid-scale standard deviation (σ ) of less than
0.3 m for the upscaled LIDAR pixel. MARE – Mean Absolute Relative Error; BC – Bhattacharya Coefficient. Here mean and
standard deviation refer to the spatial distribution, unlike the Prior mean and standard deviation used in Base-AM. Successful
retrievals are for pixels with local incidence angles in the 30º-45º range and relative residual backscatter (RRB) of less than 30%
for each of the four flights (see Table 4).

| Flight (GMT) | N Layer | Spatial SD μ [m] | | | Spatial SD σ [m] | | | MARE SD | | BC SD | |
|---|---|---|---|---|---|---|---|---|---|---|---|
| | | Retrieved | MSHM | LIDAR | Retrieved | MSHM | LIDAR | Retrieved-LIDAR | MSHM-LIDAR | Retrieved-LIDAR | MSHM-LIDAR |
| 18:11:38 | 1 | 1.41 | 1.42 | 1.40 | 0.33 | 0.18 | 0.26 | 0.19 | 0.09 | 0.90 | 0.78 |
| 18:43:20 | | 1.27 | 1.39 | 1.41 | 0.32 | 0.19 | 0.25 | 0.21 | 0.08 | 0.90 | 0.85 |
| 18:59:02 | | 1.48 | 1.38 | 1.42 | 0.37 | 0.20 | 0.25 | 0.21 | 0.07 | 0.90 | 0.82 |
| 20:23:38 | | 1.68 | 1.52 | 1.66 | 0.38 | 0.17 | 0.19 | 0.24 | 0.12 | 0.66 | 0.50 |
| 18:11:38 | 2 | 1.41 | 1.42 | 1.40 | 0.35 | 0.18 | 0.26 | 0.15 | 0.09 | 0.95 | 0.77 |
| 18:43:20 | | 1.29 | 1.39 | 1.41 | 0.32 | 0.19 | 0.25 | 0.16 | 0.08 | 0.92 | 0.85 |
| 18:59:02 | | 1.41 | 1.38 | 1.42 | 0.35 | 0.20 | 0.25 | 0.15 | 0.07 | 0.92 | 0.82 |
| 20:23:38 | | 1.67 | 1.52 | 1.66 | 0.45 | 0.17 | 0.19 | 0.22 | 0.12 | 0.76 | 0.50 |




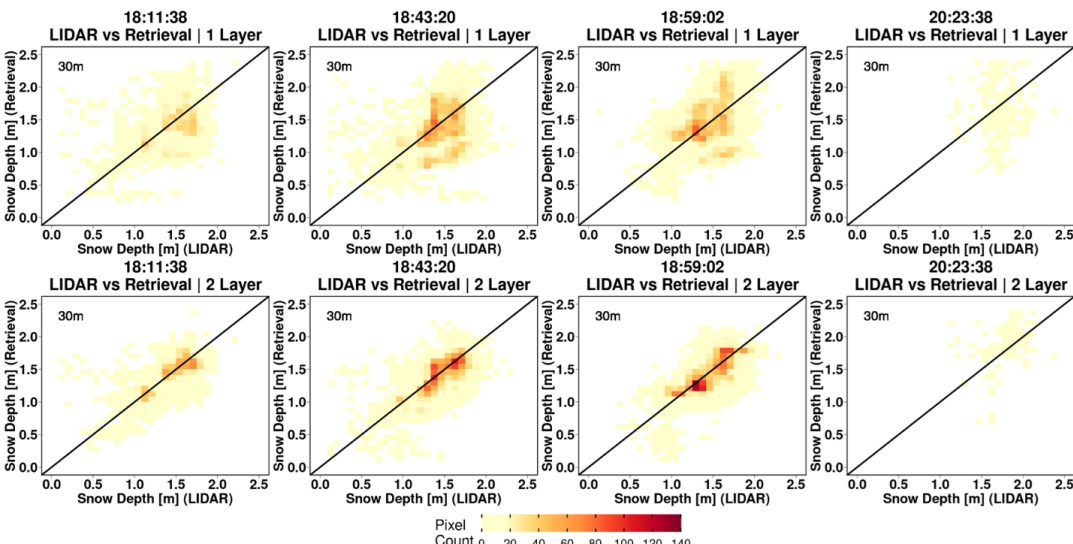

**Figure 12:** Heatmap of LIDAR versus successful snow depth retrievals at 30 m resolution using overlapping LIDAR and retrieval pixels. Successful retrievals are for pixels with local SnowSAR incidence angles in the 30°- 45° range and relative residual backscatter (RRB) of less than 30% for each of the four flights (see Table 4). LIDAR SD in pixels with subgrid scale variability corresponding to standard deviation of less than 0.3 m for the upscaled 90 m LIDAR pixel are not included.





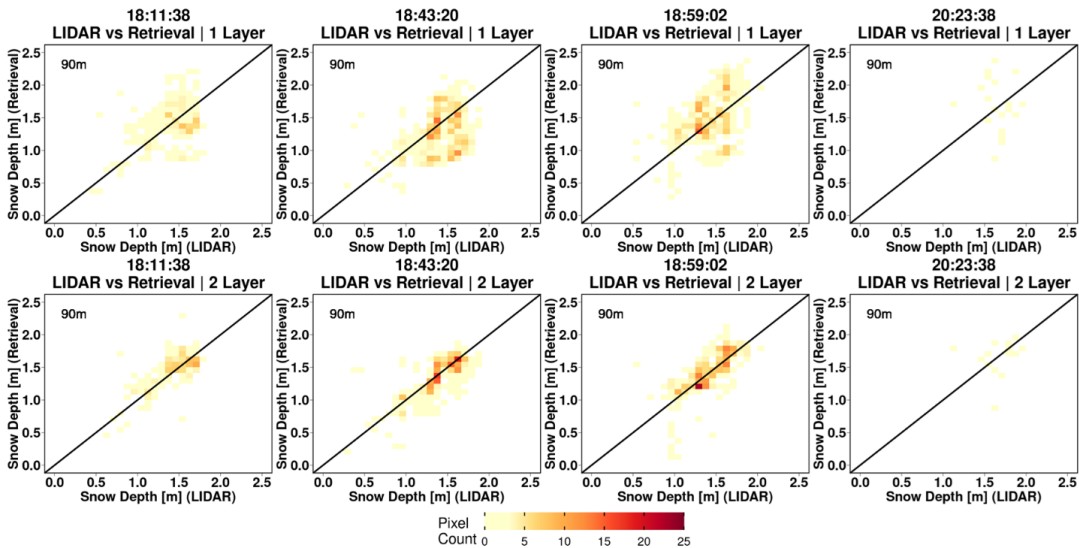

**Figure 13:** Heatmaps of LIDAR versus successful snow depth retrievals at 90 m resolution using overlapping LIDAR and retrieval pixels. Successful retrievals are for pixels with local SnowSAR incidence angles in the 30º- 45º range and relative residual backscatter (RRB) of less than 30% for each of the four flights (see Table 4). LIDAR SD in pixels with subgrid scale variability corresponding to standard deviation of less than 0.3 m for the upscaled 90 m LIDAR pixel are not included.

Composite spatial maps of successful SWE retrievals from all flights overlain by the snowpit measurements between 20-24 February are shown in Fig. 14. Note the consistency at 30 m and 90 m resolutions as well as the overall agreement between SWE at snowpits and SWE retrievals on the flightlines. Tables 7 and 8 summarize the average absolute relative errors between snowpits and all SWE retrievals within 100 m of the snowpits. The results are significantly better for two-layer snowpack retrievals. The mean absolute relative errors at 30 m resolution are 0.22 and 0.13 for 1 layer and 2 layer snowpacks respectively. The mean absolute relative errors at 90 m resolution are 0.2 and 0.12 for 1 layer and 2 layer snowpacks respectively. There is a variable number of pixels for each snow pit, which in the case of 51S is so small that indicates it is not in the flight path. After removing snowpits in the central area marked in Fig. A12 is due to very heterogeneous landcover including water, forest, (4500) and proximity to roads (53W and 44E), the average absolute relative SWE residuals is 5-7% (15-18%) for the two-layer (single-layer) retrieval algorithm.



543

544

**Table 7:** Evaluation of successful SWE retrievals at 30 m resolution against SWE at SnowEx'17 snow pits and retrieved snowpacks at 30 m resolution. All N pixels with centroids within 100 m of each snow pit are in the Grasslands (according to the Landcover dataset at 30 m resolution, see Table 1). SD – snow depth. Shaded rows correspond to large local MARE (Mean Absolute Relative Error, Eq. 6).

| Date | x | y | Pit SD (m) | Pit SWE (m) | Retrieved SWE (m) | | MARE | | N pixels | Avg. Dist (m) | Pit ID |
|---|---|---|---|---|---|---|---|---|---|---|---|
| | | | | | 1 Lyr | 2 Lyr | 1 Lyr | 2 Lyr | | | |
| 2/20/2017 | -108.184 | 39.014 | 1.15 | 0.368 | 0.455 | 0.386 | 0.236 | 0.049 | 28 | 18 | KC1C |
| 2/20/2017 | -108.184 | 39.014 | 1.19 | 0.386 | 0.457 | 0.387 | 0.184 | 0.003 | 27 | 12 | KC1E |
| 2/20/2017 | -108.184 | 39.014 | 1.18 | 0.386 | 0.456 | 0.387 | 0.181 | 0.003 | 26 | 15 | KC1N |
| 2/20/2017 | -108.184 | 39.013 | 1.24 | 0.414 | 0.456 | 0.387 | 0.101 | 0.065 | 27 | 20 | KC1S |
| 2/20/2017 | -108.184 | 39.014 | 1.30 | 0.435 | 0.455 | 0.385 | 0.046 | 0.115 | 29 | 11 | KC1W |
| 2/22/2017 | -108.136 | 39.006 | 1.32 | 0.436 | 0.556 | 0.484 | 0.275 | 0.110 | 22 | 8 | 29E |
| 2/22/2017 | -108.090 | 39.021 | 1.65 | 0.583 | 0.685 | 0.596 | 0.175 | 0.022 | 19 | 17 | 38E |
| *2/22/2017* | *-108.060* | *39.030* | *2.10* | *0.763* | *0.368* | *0.449* | *0.518* | *0.412* | *12* | *16* | *53W* |
| 2/22/2017 | -108.044 | 39.017 | 1.68 | 0.566 | 0.480 | 0.505 | 0.152 | 0.108 | 5 | 51 | 63E |
| 2/22/2017 | -108.049 | 39.017 | 1.49 | 0.48 | 0.494 | 0.513 | 0.029 | 0.069 | 13 | 29 | 63W |
| 2/22/2017 | -108.029 | 39.032 | 1.66 | 0.55 | 0.558 | 0.581 | 0.015 | 0.056 | 18 | 15 | 67N |
| *2/23/2017* | *-108.067* | *39.029* | *2.13* | *0.761* | *0.593* | *0.504* | *0.221* | *0.338* | *9* | *23* | *44E* |
| *2/23/2017* | *-108.061* | *39.030* | *1.59* | *0.568* | *0.365* | *0.408* | *0.357* | *0.282* | *3* | *75* | *51S* |
| 2/24/2017 | -108.033 | 39.030 | 1.80 | 0.576 | 0.657 | 0.573 | 0.141 | 0.005 | 20 | 10 | 0 |
| 2/24/2017 | -108.033 | 39.030 | 1.84 | 0.598 | 0.652 | 0.581 | 0.090 | 0.028 | 21 | 14 | 800 |
| 2/24/2017 | -108.033 | 39.030 | 1.80 | 0.571 | 0.650 | 0.581 | 0.138 | 0.018 | 22 | 19 | 1390 |
| 2/24/2017 | -108.033 | 39.030 | 1.75 | 0.566 | 0.654 | 0.581 | 0.155 | 0.027 | 21 | 15 | 2000 |
| 2/24/2017 | -108.033 | 39.030 | 1.67 | 0.560 | 0.654 | 0.581 | 0.168 | 0.037 | 21 | 9 | 2500 |
| *2/24/2017* | *-108.034* | *39.030* | *1.12* | *0.331* | *0.660* | *0.580* | *0.994* | *0.752* | *18* | *19* | *4500* |




















**Table 8**: Evaluation of successful SWE retrievals at 90 m resolution against SWE at SnowEx'17 snow pits and retrieved snowpacks at 90 m resolution. All N pixels with centroids within 100 m of each snow pit are in the Grasslands (according to the Landcover dataset at 90 m resolution, see Table 1). SD – Snow depth. Rows in italics correspond to large local MARE (Mean Absolute Relative Error, Eq. 6).

| Date | x | y | Pit SD (m) | Pit SWE (m) | Retrieved SWE (m) | | Mean Abs Rel Error | | N pixels | Avg. Dist (m) | Pit ID |
|---|---|---|---|---|---|---|---|---|---|---|---|
| | | | | | 1 Lyr | 2 Lyr | 1 Lyr | 2 Lyr | | | |
| 2/20/2017 | -108.184 | 39.014 | 1.15 | 0.368 | 0.473 | 0.398 | 0.29 | 0.08 | 4 | 18 | KC1C |
| 2/20/2017 | -108.184 | 39.014 | 1.19 | 0.386 | 0.471 | 0.397 | 0.22 | 0.03 | 3 | 12 | KC1E |
| 2/20/2017 | -108.184 | 39.014 | 1.18 | 0.386 | 0.473 | 0.399 | 0.22 | 0.03 | 2 | 29 | KC1N |
| 2/20/2017 | -108.184 | 39.013 | 1.24 | 0.414 | 0.474 | 0.398 | 0.15 | 0.04 | 3 | 27 | KC1S |
| 2/20/2017 | -108.184 | 39.014 | 1.3 | 0.435 | 0.476 | 0.399 | 0.09 | 0.08 | 3 | 47 | KC1W |
| 2/22/2017 | -108.136 | 39.006 | 1.32 | 0.436 | 0.572 | 0.490 | 0.31 | 0.12 | 2 | 39 | 29E |
| *2/22/2017* | *-108.060* | *39.030* | *2.10* | *0.763* | *0.340* | *0.384* | *0.55* | *0.50* | *1* | *43* | *53W* |
| 2/22/2017 | -108.044 | 39.017 | 1.68 | 0.566 | 0.454 | 0.499 | 0.20 | 0.12 | 1 | 75 | 63E |
| 2/22/2017 | -108.049 | 39.017 | 1.49 | 0.480 | 0.521 | 0.530 | 0.09 | 0.10 | 1 | 29 | 63W |
| 2/22/2017 | -108.029 | 39.032 | 1.66 | 0.550 | 0.529 | 0.553 | 0.04 | 0.01 | 4 | 47 | 67N |
| 2/23/2017 | -108.067 | 39.029 | 2.13 | 0.761 | 0.751 | 0.606 | 0.01 | 0.20 | 1 | 70 | 44E |
| 2/24/2017 | -108.033 | 39.030 | 1.8 | 0.576 | 0.718 | 0.601 | 0.25 | 0.04 | 3 | 60 | 0 |
| 2/24/2017 | -108.033 | 39.030 | 1.84 | 0.598 | 0.717 | 0.600 | 0.20 | 0.00 | 2 | 57 | 800 |
| 2/24/2017 | -108.033 | 39.030 | 1.80 | 0.571 | 0.717 | 0.600 | 0.26 | 0.05 | 2 | 55 | 1390 |
| 2/24/2017 | -108.033 | 39.030 | 1.75 | 0.566 | 0.687 | 0.592 | 0.21 | 0.05 | 1 | 54 | 2000 |
| 2/24/2017 | -108.033 | 39.030 | 1.67 | 0.560 | 0.687 | 0.592 | 0.23 | 0.06 | 1 | 54 | 2500 |
| *2/24/2017* | *-108.034* | *39.030* | *1.12* | *0.331* | *0.687* | *0.592* | *1.08* | *0.79* | *1* | *62* | *4500* |
| 2/20/2017 | -108.184 | 39.014 | 1.15 | 0.368 | 0.473 | 0.398 | 0.29 | 0.08 | 4 | 18 | KC1C |
| 2/20/2017 | -108.184 | 39.014 | 1.19 | 0.386 | 0.471 | 0.397 | 0.22 | 0.03 | 3 | 12 | KC1E |

568

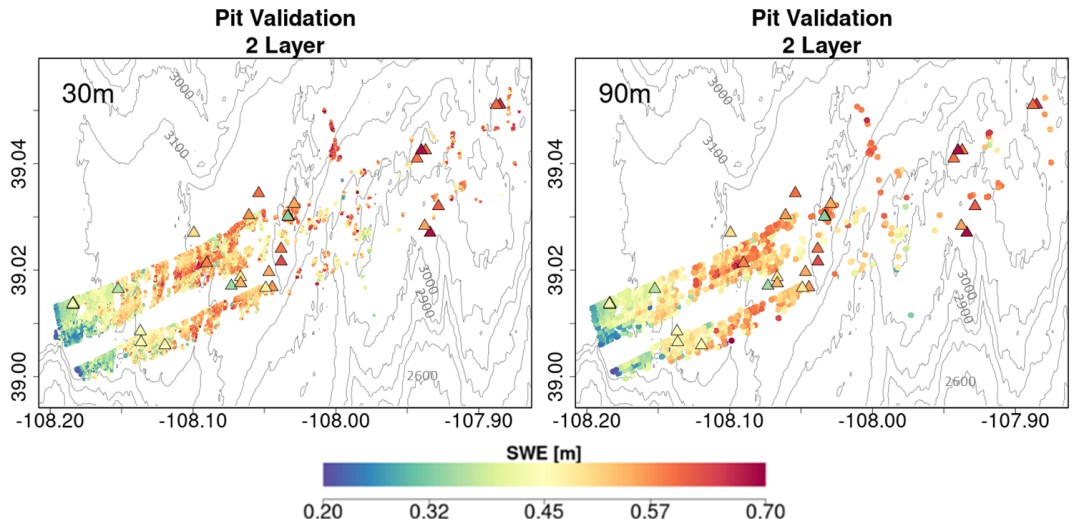

569

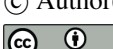



**Figure 14:** Composite spatial distribution of SWE (2-layer snowpack) successfully retrieved at 30m (left) and 90m (right) resolution for grassland pixels for the four SnowSAR flights. Snow pits (19-24 Feb, Fig. 4, Tables 7 and 8) are marked by triangles colored according to SWE. SnowEx'17 snow pit locations are marked by triangles and colored according to SWE. Successful retrievals are for pixels with local incidence angles in the 30º- 45º range and relative residual backscatter (RRB) of less than 30% for each of the four flights (see Table 4). Gray elevation contours are plotted every 100m.

## 6. Conclusion

A Bayesian physical-statistical SWE retrieval framework leveraging prior work (CB20, CB22, P17, P23, Fig. 5) was applied to airborne X- and Ku-band measurements yielding robust results for multiple flights including overlapping paths over grassland and mixed grassland and forest in Grand Mesa, Colorado. Prior distributions of snowpack parameters were obtained from a multilayer snow hydrology model with atmospheric forcing derived from operational NWP forecasts and analysis (CB20, CB22). In order to reconcile the number of independent measurements, physical constraints, and reduce the number of snowpack parameters, snowpack stratigraphy was mapped into single-layer and two-layer snowpacks for Bayesian inference using Base-AM (P17, P23). The SnowSAR measurements were averaged to 30 and 90 m resolutions, and retrievals were conducted for every measurement pixel along the flight lines. Retrievals for measurements with convergence backscatter relative errors less than 30% (±1.2dB) and for incidence angles in the 30º-45º range were considered successful over grasslands, corresponding to 75-87% of all retrievals.

The retrievals (i.e. the local mean of the posterior distribution) were evaluated against the spatial distribution of LIDAR snow depth estimates up to 2 m and against snowpit SWE measurements up to 700 mm. Note that the LIDAR and snowpit measurements are not at the same time of the SnowSAR flights, and the assessment of retrieval skill was conducted over a period of five days without snowfall or significant day-to-day weather changes. The two-layer snowpack retrievals perform better overall being able to capture the statistical variability of snow depth with high fidelity, with SWE relative errors ≤ 7% as compared with 18% for single-layer SWE retrievals, and snow depth absolute retrieval residuals 10-20% depending on landcover heterogeneity and measurement uncertainty. The statistical structure of retrieved snow depth is similar to that estimated by LIDAR, which is indicative of the retrievals ability to capture snow patterns and scaling behavior to support process studies. For satellite-based monitoring from space in the context of a future snow mission, time-series of measurements would be available that should improve the estimates of the priors for the present retrieval cycle. This is not possible for field experiments such as SnowEx'17, and thus improved results would be expected under realistic satellite-based applications. NWP forecasts are available worldwide and therefore this retrieval framework can be applied to SAR measurements anywhere.

The radar model used in this study (MEMLS) does incorporate snow-ground-vegetation scattering interactions. Grassland vegetation during the accumulation season is assumed to be submerged and the impact of vegetation is included in the estimation of the background backscatter ($\sigma_{bkg}$, Fig. 1). Because the landcover data are categorical, in addition to the uncertainty of the data at 30 m resolution, additional uncertainty is tied to the selection of homogeneous grassland pixels at 90 resolution, which explains some of the unsuccessful retrievals especially along the grassland-forest




and shrub boundaries. The potential for estimating $\sigma_{bkg}$ independently for each location as
proposed by Cao and Barros (2023) provides an alternative to simplify the retrieval workflow and
target the Bayesian inference to the snowmass and volume backscatter ( $\sigma_{vol}=\sigma_{total}-\sigma_{bkg}$).
Airborne measurements are characterized by large changes in viewing geometry across the flight-
line and due to other factors such as variable winds and turbulence depending on weather
conditions, thus pointing to improved skill from satellite platforms. Building on previous mission
concepts (e.g. Rott et al. 2012) and leveraging substantial theory advances and field campaigns in
the last decade, this study demonstrates the utility and effectiveness of X-and Ku-band SAR
technology to remotely monitor snowmass at high spatial resolution and with accuracy and
uncertainty that meet the requirements expressed in the most recent Earth Science and Applications
from Space Decadal Survey (NASEM, 2018).

**7. Appendix A**

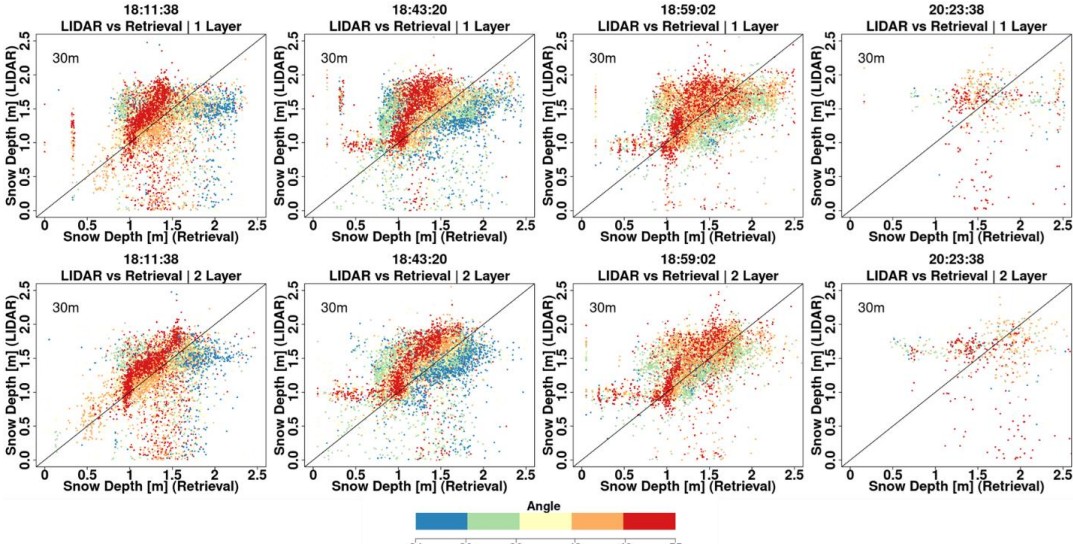


**Figure A1**: Same as Fig. 7b with pixels color coded according to the local SnowSAR incidence angle for all four flightlines and
for single-(top row) and two-layer (bottom row) retrievals at 30 m resolution.



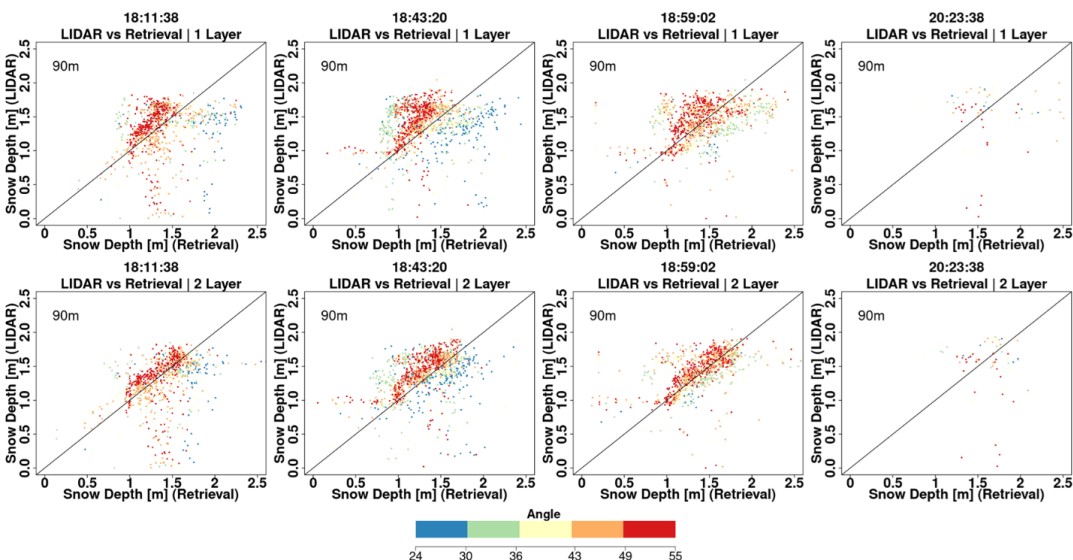

**Figure A2**: Same as Fig. 7b with pixels color coded according to the local SnowSAR incidence angle for all four flightlines and for single-(top row) and two-layer (bottom row) retrievals at 90 m resolution.

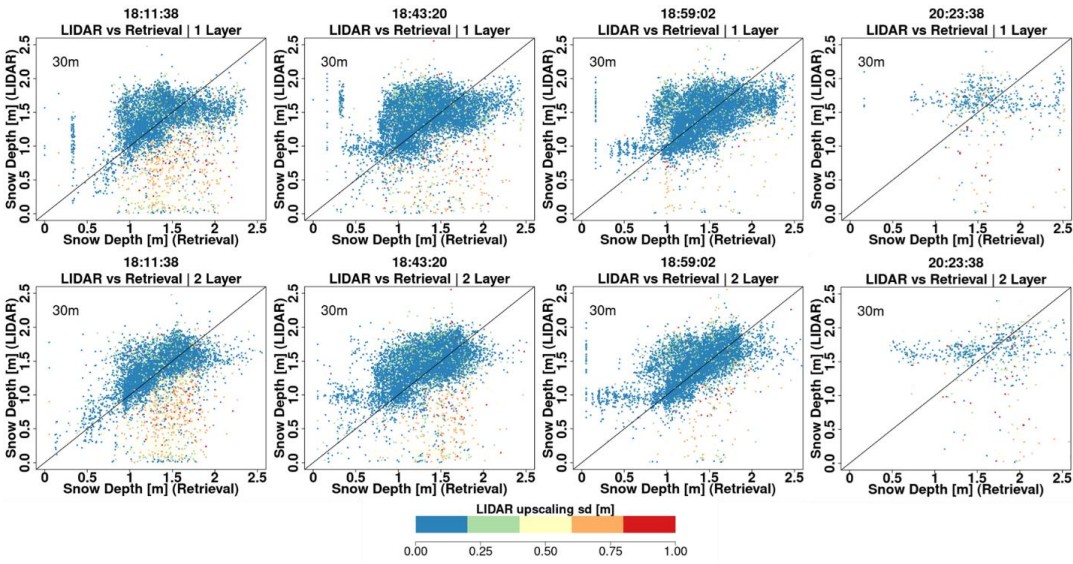

**Figure A3**: Comparison between LIDAR snow depth and successful retrievals for single and two-layer algorithms. The pixels are color coded according to the subgrid scale variability of the 30 m upscaled LIDAR pixel.




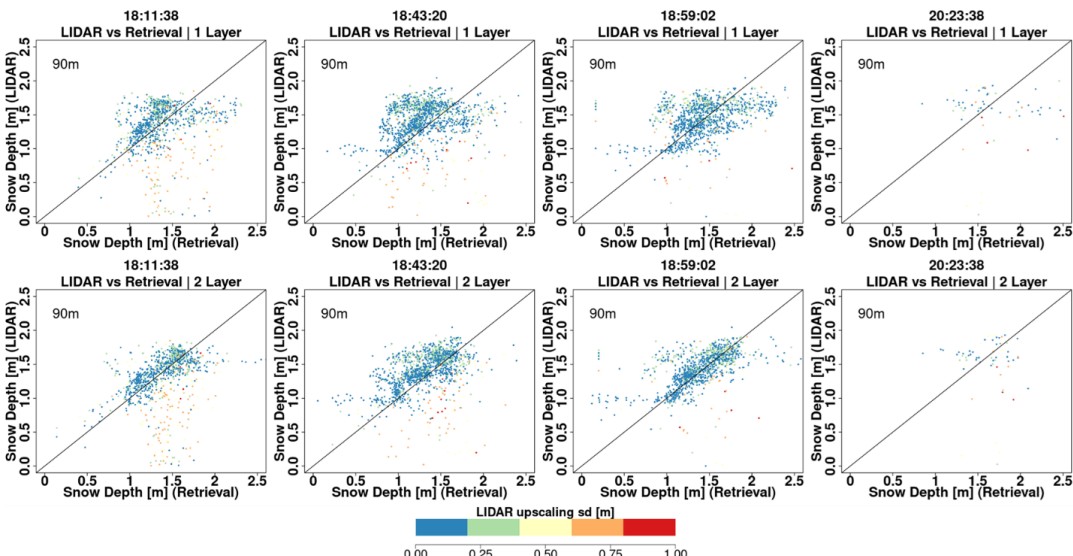


**Figure A4:** Comparison between SnowSAR snow depth and successful retrievals. The pixels are color coded according to the
subgrid scale variability of the 90 m upscaled LIDAR pixel.


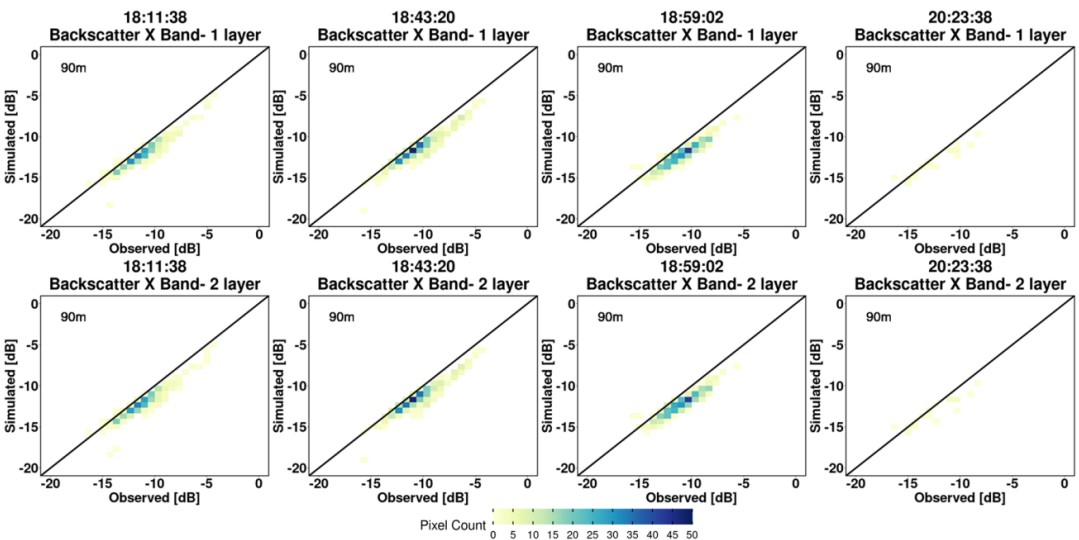


**Figure A5**: Comparison between SnowSAR backscatter (X-band) and BASE-AM converged backscatter at 90 m resolution for
successful retrievals.  Error statistics can be found in Table 4.






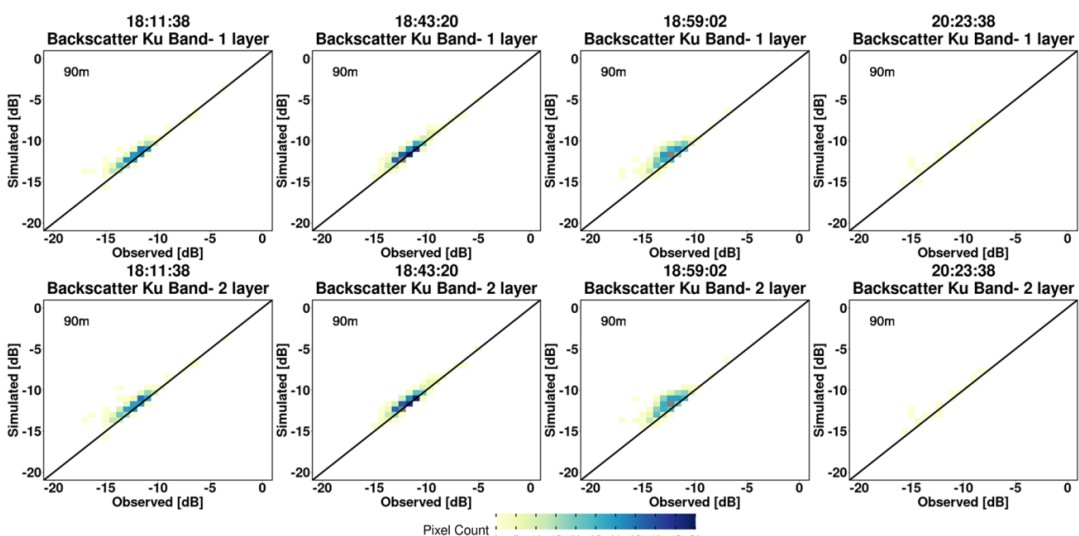

**Figure A6** - Comparison between SnowSAR backscatter (Ku-band) and Base-AM converged backscatter at 90 m resolution for successful retrievals. Error statistics can be found in Table 4.

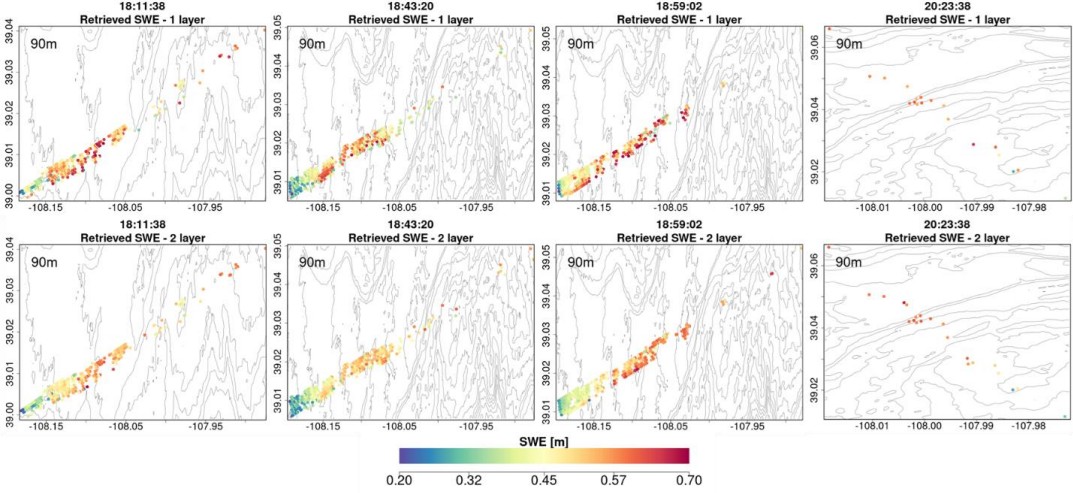

**Figure A7:** Spatial distribution of successful SWE retrievals for single- and 2-layer snowpacks at 90 m resolution. The retrievals are for grassland pixels only.



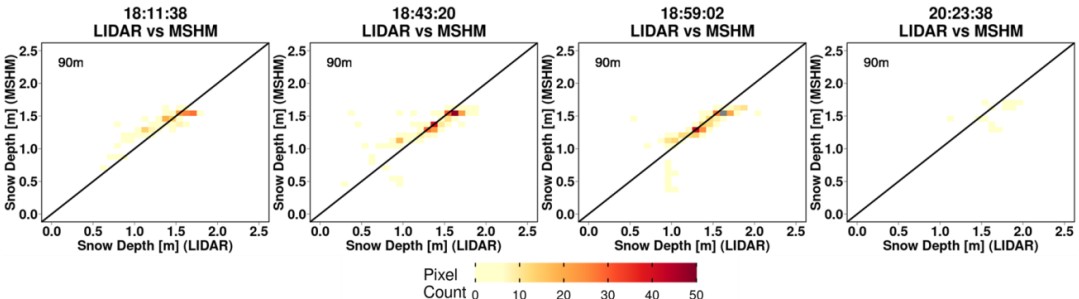

**Figure A8:** Heatmaps of LIDAR snow depth and snow depth predicted by MSHM at the time of SnowSAR flights for overlapping pixels at 90 m resolution.

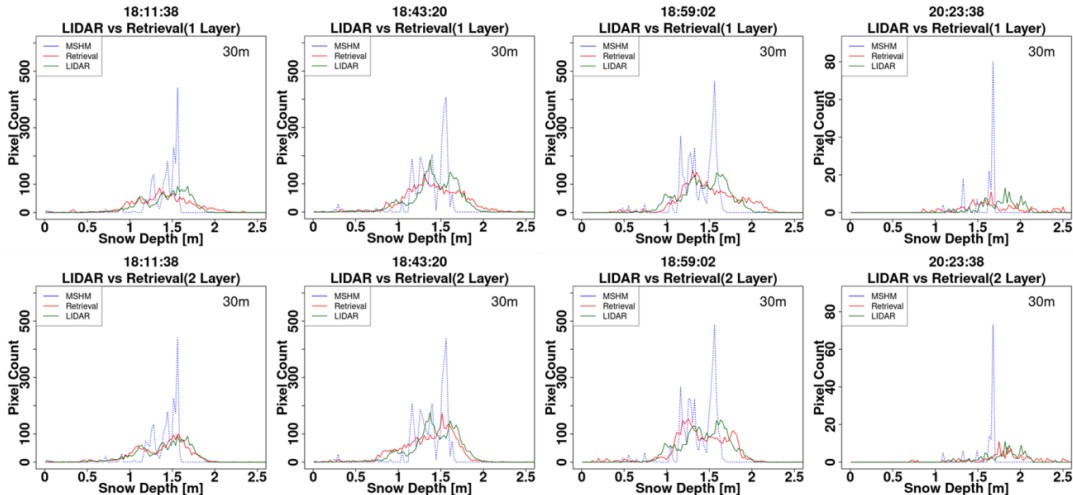

**Figure A9:** Histogram of snow depth (SD) from LIDAR, MSHM, and successful retrievals at 30 m using 1- and 2- layer snowpacks. The total number of pixels for each snow depth product is the same. Successful retrievals are for pixels with local incidence angles in the 30°- 45° range and relative residual backscatter (RRB) of less than 30% for each of the four flights (see Table 4). LIDAR SD in pixels with subgrid scale variability corresponding to standard deviation of less than 0.3 m for the upscaled 90 m LIDAR pixel are not included.



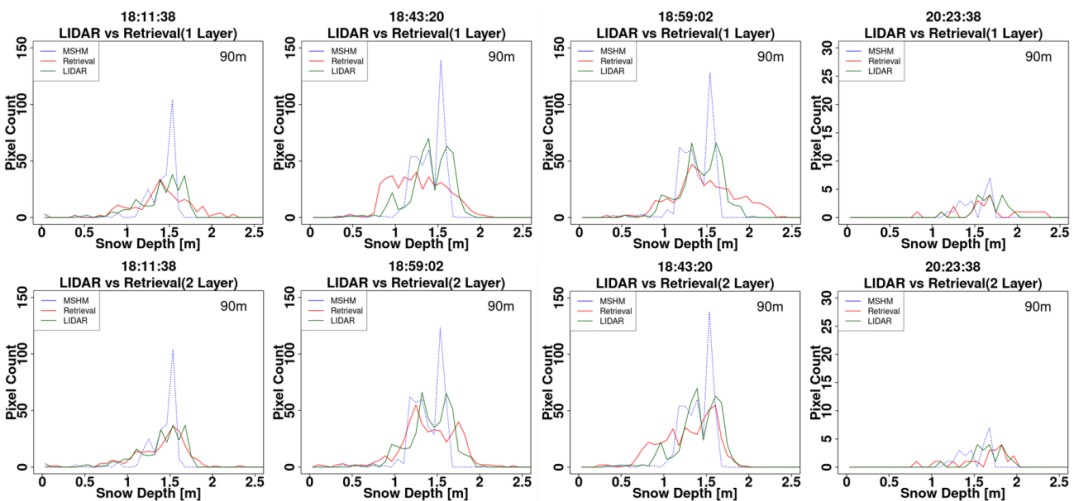

**Figure A10 -** Histogram of snow depth (SD) from LIDAR, MSHM, and successful retrievals at 90 m using 1- and 2- layer snowpacks. The total number of pixels for each snow depth product is the same. Successful retrievals are for pixels with local incidence angles in the 30°- 45° range and relative residual backscatter (RRB) of less than 30% for each of the four flights (see Table 4). LIDAR SD in pixels with subgrid scale variability corresponding to standard deviation of less than 0.3 m for the upscaled 90 m LIDAR pixel are not included.

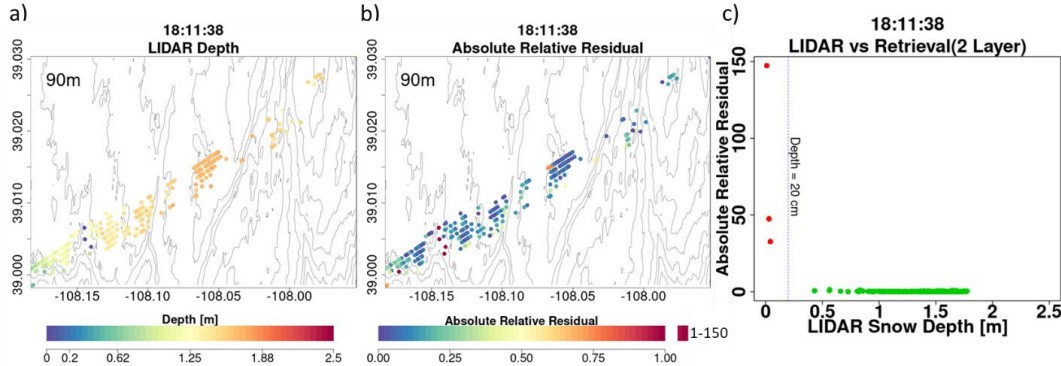

**Figure A11** - Analysis of unsuccessful retrievals for pixels with large mean snow depth residuals at 90 m resolution: a) Map of LIDAR snow depth highlighting in deep blue the locations where very shallow snow is attributed to measurement error. b) Note spatial agreement between shallow snow depth and very large residuals. c)There are only a few points at the edges of forests and shallow snow depths that are flagged not successful. The gray elevation contours are plotted every 50 m.



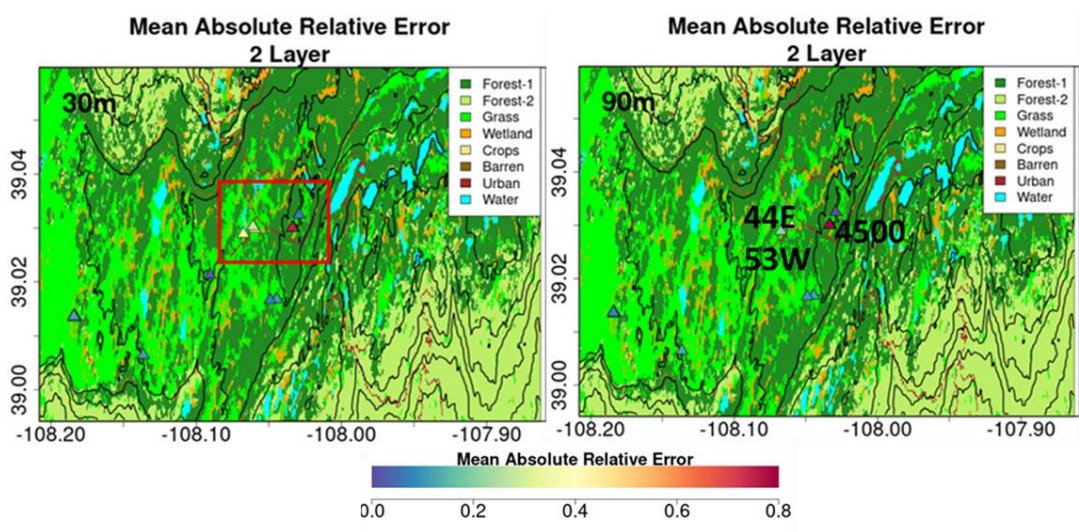

**Figure A12** – Spatial context for snow pits with very large absolute relative errors (MARE) calculated as the mean of the relative difference between SWE retrievals within 100 m of the snow pit and the values at the snow pit  Locations with very large errors (orange to red) are inside the red box marked in top plot. Snowpit 4500 is  a region of complex land cover including evergreen forest,  a road and a pond.  Snowpits 53W and 44E are close to each other on the same side of the road in expansive grassland.

## 8. Competing Interests

The contact author has declared that none of the authors has any competing interests

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
