# Peer review of "BAYESIAN PHYSICAL-STATISTICAL RETRIEVAL OF SWE AND SNOW DEPTH FROM X"

_EGUsphere, 2023_

## Referee Comment (RC1)

Snow Water Equivalent (SWE) is a key parameter in hydrological, climatological and meteorological applications. New efforts for spaceborne radar-based SWE retrieval algorithms are under development and this paper shows the capability of such retrievals using snow-physical model driven by meteo data, radiative transfer and Bayesian inference. This paper focuses on the SWE retrieval framework leveraging previous work. The paper shows the readiness and capabilities of combining existing models and products to produce a SWE retrieval for RADAR data. The paper is well constructed and provides great results for SWE retrievals using Ku band radar. The method is repeatable elsewhere and estimating the background with X-band is clever.

Refining and thinning the results section would help clarify the take home message. Most of the figures are almost duplicates and I'm not sure I see the benefit in most cases. Or it is not well explained in the text. I have specific comments throughout the paper.

Specific comments:

Line 73-75 : This is a key sentence in setting the objective but it's complicated to understand. I suggest reformulating.

Line 76: Do you need all the verb? I feel like *propose* and *evaluate* were enough. Previous studies already implemented and demonstrated.

Line 83: temporal variability relates to temporal resolution of the orbit or revisit time of the satellite. I suggest removing or adding high temporal resolution higher.

Line 86 : "a coupled multi-layer snow hydrology model"? add model

Line 127 and line 130: Should it be y not x for the retrieved variable?

Line 148 and Line 150: do we maximize P(n|y) or P(y|n)?

Line 164 : I suggest a bit more detail in the paragraph. Is y the snow depth, SWE or backscatter? Did you use the likelihood ratio to iterate in the MCMC like Pan et al?

Line 246. Wrong figure number. Should be 6.

Figure 5 : Is this the layer index? What does 15 layers mean? Top or bottom? Height or normalized height would be better. Put density on the x-axis.

Line 267: Why this value? Can you explain more this parameter. "This is an empirical factor that …"

Line 271: replace microphysics for microstructure.

Line 272: add the symbol ($l_{ex}$) that represent the correlation length in MEMLS.

Table 2: Relates to the previous comment. Why use D?. D is the equivalent grain size used in DMRT. Replace D for $l_{ex}$.

Line 298: "for each layers"

Figure 6: The figure could be clearer. Later you refer to steps, but no steps are indicated in the figure. Pretty hard to understand even if you know Bayesian SWE retrieval. This could improve the understanding of the reader.

Line 317: It is not clear how the background is estimated. Maybe Specified that the volume is modelled from MSHM in the text not just in figure 6. Then explain why only using X band. It might not be obvious to someone not familiar with the fact that X band is more sensitive to the background than Ku.

Line 358 : "toweak" change "to a weak"

Figure 8 : I don't' see the point of having 4 columns in the figure and then another figure for Ku. Can your aggregate heatmap for all dates and have x and ku in the same figure? They all look the same. I think we get the idea that the backscatter converges to the observed.

Line 471: remove in after between.

Line 475: add a coma after "In all cases".

Table 6: change title to " Same as Table 5 but for resolution = 90 m".

Line 529 – 530: Can you add the mean values stated here in the table as the last row?

Table 5, 6, 7 and 9: Any benefit in having the same table but with different resolutions? We get the idea in Table 4 regarding the resolution. Just make the point you want and move on. There is no point in having both figure 12 and 13. Just show one. I don't see any big conclusion regarding the resolution so there is no point in adding extra figures and tables.

---

## Referee Comment (RC2)

General Comments

The authors of this manuscript use a Bayesian physical-statistical model to retrieve snow depth using data that was acquired during SnowEx'17, including pit data, ASO lidar acquisitions and the X- and Ku-band SnowSAR data. The methods presented by the authors are clearly defined, with encouraging results with error ranges of about 20cm SWE. The paper is methodologically and analytically sound, but does lack context of why certain methods are being employed, and the representation of results in figures need to be improved. For instance, in the Introduction it was overall well written to describe the need for remote sensing of snow and how SnowEx'17 is an attempt to address this concern. I do feel that the end of this section needs to be improved to detail to the reader what the specific objectives of this manuscript are. For instance, the authors provide an indication that they are looking to test a physical-statistical framework to derive SWE – but what are the objectives that can be completed to evaluate this?

The paper is very detailed in its methods of preparing the data for the model, and the modeling itself. What I think is lacking here is the reason that you are completing some of these steps. For instance, in section 4.1.1. you discuss in detail how you divide the snowpack with multiple layers into a snowpack with 1 or 2 layers, but never state why a 2 layer pack would be useful (that snowpack generally has a wind slab and depth hoar layer).. The justification for many of the steps in this paper need to have a bit better context provided.

Overall the paper is of publication quality in terms of its research, but the presentation could be improved, with my general and specific comments provided here.

**Section 2. Previous Work**

Line 55: "Time-series observations are available presently from tower measurements, albeit at the point scale of the tower footprint". *I think I know what study/setup you're referring to here, but you have not referenced the papers that have been published based on them. Also, a sentence that states why you are not using a tower approach in this work would be useful.*

Line 86: "demonstrated the utility of a couple multi-layer snow hydrology coupled with a.." – *do you mean "snow hydrology model coupled…"?*

Line 133-134: "the second is the prior of the backscatter… the prior of the snowpack physical…" *this sentence is a little unclear, please revise.*

Line 144: "assuming that we have good understanding" *grammar issue here "assuming that we have a good understanding".*

Table 1: I may have missed it, but why are all the datasets being upscaled to 90m? I also noticed this discussed in lines 175-176, but there was no justification as to why – please include.

Line 299: "from the multilayer snowpack simulated by MSHM as for the single layer case" *I'm not sure what you are trying to say here*.

Line 366: "restructingthe" – *restricting the*

Figure 7 – None of the panels have a letter denoting which panel they are. The panels with the red box should be made into a new figure that zooms into the distribution of values within the red box – there are discussions in the Results section of what's happening here, but it is difficult for the reader to confirm what the text is saying graphically because it is far too small.

Figure 8, 9, 10, 11 also has no panel labels.

Line 433: "Fig. A7" – where is A7?

Line 445: "Fig A8" – where is A8? Be specific when referring to the appendix.

Line 477: "Fig A11" – where?

Line 534: "including water, forest (4500) and proximity" – *what does 4500 refer to?*

Table 7: *There is a note in the caption that "shaded rows correspond to large local MARE" – however there are no shaded rows in the table.*

---

## Author Comment (AC2)

Snow Water Equivalent (SWE) is a key parameter in hydrological, climatological and meteorological applications. New efforts for spaceborne radar-based SWE retrieval algorithms are under development and this paper shows the capability of such retrievals using snow-physical model driven by meteo data, radiative transfer and Bayesian inference. This paper focuses on the SWE retrieval framework leveraging previous work. The paper shows the readiness and capabilities of combining existing models and products to produce a SWE retrieval for RADAR data. The paper is well constructed and provides great results for SWE retrievals using Ku band radar. The method is repeatable elsewhere and estimating the background with X-band is clever.

Refining and thinning the results section would help clarify the take home message. Most of the figures are almost duplicates and I'm not sure I see the benefit in most cases. Or it is not well explained in the text. I have specific comments throughout the paper.

Specific comments:

Line 73-75 : This is a key sentence in setting the objective but it's complicated to understand. I suggest reformulating.

Line 76: Do you need all the verb? I feel like *propose* and *evaluate* were enough. Previous studies already implemented and demonstrated.

Line 83: temporal variability relates to temporal resolution of the orbit or revisit time of the satellite. I suggest removing or adding high temporal resolution higher.

Line 86 : "a coupled multi-layer snow hydrology model"? add model

Line 127 and line 130: Should it be y not x for the retrieved variable?

Line 148 and Line 150: do we maximize P(n|y) or P(y|n)?

Line 164 : I suggest a bit more detail in the paragraph. Is y the snow depth, SWE or backscatter? Did you use the likelihood ratio to iterate in the MCMC like Pan et al?

Line 246. Wrong figure number. Should be 6.

Figure 5 : Is this the layer index? What does 15 layers mean? Top or bottom? Height or normalized height would be better. Put density on the x-axis.

Line 267: Why this value? Can you explain more this parameter. "This is an empirical factor that …"

Line 271: replace microphysics for microstructure.

Line 272: add the symbol ($l_{ex}$) that represent the correlation length in MEMLS.

Table 2: Relates to the previous comment. Why use D?. D is the equivalent grain size used in DMRT. Replace D for $l_{ex}$.

Line 298: "for each layers"

Figure 6: The figure could be clearer. Later you refer to steps, but no steps are indicated in the figure. Pretty hard to understand even if you know Bayesian SWE retrieval. This could improve the understanding of the reader.

Line 317: It is not clear how the background is estimated. Maybe Specified that the volume is modelled from MSHM in the text not just in figure 6. Then explain why only using X band. It might not be obvious to someone not familiar with the fact that X band is more sensitive to the background than Ku.

Line 358 : "toweak" change "to a weak"

Figure 8 : I don't' see the point of having 4 columns in the figure and then another figure for Ku. Can your aggregate heatmap for all dates and have x and ku in the same figure? They all look the same. I think we get the idea that the backscatter converges to the observed.

Line 471: remove in after between.

Line 475: add a coma after "In all cases".

Table 6: change title to " Same as Table 5 but for resolution = 90 m".

Line 529 – 530: Can you add the mean values stated here in the table as the last row?

Table 5, 6, 7 and 9: Any benefit in having the same table but with different resolutions? We get the idea in Table 4 regarding the resolution. Just make the point you want and move on. There is no point in having both figure 12 and 13. Just show one. I don't see any big conclusion regarding the resolution so there is no point in adding extra figures and tables.

**Reply To Reviewer 1**

We thank the Reviewer for the helpful comments and suggestions. The Reviewer's comments are in black. Our replies are in blue.

Snow Water Equivalent (SWE) is a key parameter in hydrological, climatological and meteorological applications. New efforts for spaceborne radar-based SWE retrieval algorithms are under development and this paper shows the capability of such retrievals using snow-physical model driven by meteo data, radiative transfer and Bayesian inference. This paper focuses on the SWE retrieval framework leveraging previous work. The paper shows the readiness and capabilities of combining existing models and products to produce a SWE retrieval for RADAR data. The paper is well constructed and provides great results for SWE retrievals using Ku band radar. The method is repeatable elsewhere and estimating the background with X-band is clever.

Thank you.

Refining and thinning the results section would help clarify the take home message. Most of the figures are almost duplicates and I'm not sure I see the benefit in most cases. Or it is not well explained in the text. I have specific comments throughout the paper.

We made a conscientious effort to eliminate duplicates in the main paper and revised the manuscript for clarity.

Specific comments:

Line 73-75 : This is a key sentence in setting the objective but it's complicated to understand. I suggest reformulating.

Line 73 is Line 80 in the revised manuscript: Sentence was revised and references were added.

Line 76: Do you need all the verb? I feel like *propose* and *evaluate* were enough. Previous studies already implemented and demonstrated.

Line 76 is Line 83 in the revised manuscript: Revised.

Line 83: temporal variability relates to temporal resolution of the orbit or revisit time of the satellite. I suggest removing or adding high temporal resolution higher.

Line 83 is Line 89 in the revised manuscript: Removed the temporal reference as suggested.

Line 86 : "a coupled multi-layer snow hydrology model"? add model

Line 86 is Line 92 in the revised manuscript: Revised.

Line 127 and line 130: Should it be y not x for the retrieved variable?

Line 127 is Line 133 in the revised manuscript:  Point well taken. Section 2.2 was carefully edited for clarity.  We start with general indirect measurements D to pose the general problem, and then for a specific instrument we replace D by y.  η includes the geophysical variables x as well as the model parameters.

Line 148 and Line 150: do we maximize P(n|y) or P(y|n)?

Line 148 is Line 154 in the revised manuscript:  To maximize P(η|y) we need to maximize P(y| η) since P(η) is a prior probability.  This should be clearer after the editing.

Line 164 : I suggest a bit more detail in the paragraph. Is y the snow depth, SWE or backscatter? Did you use the likelihood ratio to iterate in the MCMC like Pan et al?

Line 164 is Line 167 in the revised manuscript:  Yes.  This was added to the text.

Line 246. Wrong figure number. Should be 6.

Line 246 is Line 272 in the revised manuscript: Yes.  Thank you.  This was corrected.

Figure 5 : Is this the layer index? What does 15 layers mean? Top or bottom? Height or normalized height would be better. Put density on the x-axis.

Figure 5 is Figure 6 in the revised manuscript: The layering scheme in MSHM is from bottom to top following the evolution of the snowpack during the accumulation.  The higher index layer is the top layer; the bottom layer is always the first layer.

We switched the axis in Figure 5 to have density in the x-axis which is a more intuitive way to visualize the density profile of the snowpack as suggested by the Reviewer.   The figure caption was also improved for clarity and detail.

Line 267: Why this value? Can you explain more this parameter. "This is an empirical factor that …"

Line 267 is Line 292 in the revised manuscript: Revised as suggested.

Line 271: replace microphysics for microstructure.

Line 271 is Line 297 in the revised manuscript: Revised as suggested.

Line 272: add the symbol ($l_{ex}$) that represent the correlation length in MEMLS.

Line 272 is Line 299 in the revised manuscript: Revised as suggested.

Table 2: Relates to the previous comment. Why use D?. D is the equivalent grain size used in DMRT. Replace D for $l_{ex}$.

There was a notation confusion between the snow grainsize and correlation length. This is fixed now.

Line 298: "for each layers"

Line 298 is Line 335 in the revised manuscript: Revised as suggested.

Figure 6: The figure could be clearer. Later you refer to steps, but no steps are indicated in the figure. Pretty hard to understand even if you know Bayesian SWE retrieval. This could improve the understanding of the reader.

Figure 6, now Figure 5, was completely revised with each step identified. We hope the workflow is clearer now.

Line 317: It is not clear how the background is estimated. Maybe Specified that the volume is modelled from MSHM in the text not just in figure 6. Then explain why only using X band. It might not be obvious to someone not familiar with the fact that X band is more sensitive to the background than Ku.

Done. This is now in Line 316. We added a reference to justify the choice of HH-pol and revised the sentence. As pointed out in Section 3, SnowSAR Ku HH-pol measurements are not reliable.

Line 358 : "toweak" change "to a weak"

Line 358 is Line 387 in the revised manuscript: Revised as suggested.

Figure 8 : I don't' see the point of having 4 columns in the figure and then another figure for Ku. Can your aggregate heatmap for all dates and have x and ku in the same figure? They all look the same. I think we get the idea that the backscatter converges to the observed.

We understand the Reviewer's point. Each heatmap synthesizes independent retrievals over different flight paths and thus for different viewing geometries. Because of the importance of showing the robustness of the algorithm, we prefer to keep the results. We did revise the figure to make it easier to read and less crowded. We hope this is acceptable.

Line 471: remove in after between.

Line 471 is Line 493 in the revised manuscript: Revised as suggested.

Line 475: add a coma after "In all cases".

Line 475 is Line 498 in the revised manuscript: Revised as suggested.

Table 6: change title to " Same as Table 5 but for resolution = 90 m".

Done.  Tabel was moved to Appendix as Table A1.

Line 529 – 530: Can you add the mean values stated here in the table as the last row?

Line 529 is Line 521 in the revised manuscript: Revised as suggested.

Table 5, 6, 7 and 9: Any benefit in having the same table but with different resolutions? We get the idea in Table 4 regarding the resolution. Just make the point you want and move on. There is no point in having both figure 12 and 13. Just show one. I don't see any big conclusion regarding the resolution so there is no point in adding extra figures and tables.

The Reviewer's point is well taken.  Figures and Tables for 90 m resolution were moved to Appendix.

Thank you.

[revised manuscript text omitted]

Temperature
Air Pressure
Incoming shortwave radiation
Incoming longwave radiation
Wind speed
Humidity
Albedo | Snow Temperature Profile
Soil Temperature Profile
Snow Density Profile
Snow Depth Layering Profile
Liquid Water Content Profile
Snow Correlation Length Profile | Cao and Barros (2020) |
| MEMLS | Snow Temperature Profile
Soil Temperature Profile
Snow Density Profile
Snow Depth Layering Profile
Snow Correlation Length Profile
Cross polarization fraction
Ground rms height | Diffused Reflectivity Profile
Specular Reflectivity Profile
Total Backscatter Coefficient | Proksch et al. (2015) |
| Base-AM | Equivalent Snow Temperature Prior
Equivalent Soil Temperature Prior
Equivalent Snow Density Prior
Equivalent Snow Depth Prior
Correlation Length
Cross polarization fraction
Ground rms height

[revised manuscript text omitted]

---

## Author Comment (AC3)

General Comments

The authors of this manuscript use a Bayesian physical-statistical model to retrieve snow depth using data that was acquired during SnowEx'17, including pit data, ASO lidar acquisitions and the X- and Ku-band SnowSAR data. The methods presented by the authors are clearly defined, with encouraging results with error ranges of about 20cm SWE. The paper is methodologically and analytically sound, but does lack context of why certain methods are being employed, and the representation of results in figures need to be improved. For instance, in the Introduction it was overall well written to describe the need for remote sensing of snow and how SnowEx'17 is an attempt to address this concern. I do feel that the end of this section needs to be improved to detail to the reader what the specific objectives of this manuscript are. For instance, the authors provide an indication that they are looking to test a physical-statistical framework to derive SWE – but what are the objectives that can be completed to evaluate this?

The paper is very detailed in its methods of preparing the data for the model, and the modeling itself. What I think is lacking here is the reason that you are completing some of these steps. For instance, in section 4.1.1. you discuss in detail how you divide the snowpack with multiple layers into a snowpack with 1 or 2 layers, but never state why a 2 layer pack would be useful (that snowpack generally has a wind slab and depth hoar layer).. The justification for many of the steps in this paper need to have a bit better context provided.

Overall the paper is of publication quality in terms of its research, but the presentation could be improved, with my general and specific comments provided here.

**Section 2. Previous Work**

Line 55: "Time-series observations are available presently from tower measurements, albeit at the point scale of the tower footprint". *I think I know what study/setup you're referring to here, but you have not referenced the papers that have been published based on them. Also, a sentence that states why you are not using a tower approach in this work would be useful.*

Line 86: "demonstrated the utility of a couple multi-layer snow hydrology coupled with a.." – *do you mean "snow hydrology model coupled…"?*

Line 133-134: "the second is the prior of the backscatter… the prior of the snowpack physical…" *this sentence is a little unclear, please revise.*

Line 144: "assuming that we have good understanding" *grammar issue here "assuming that we have a good understanding".*

Table 1: I may have missed it, but why are all the datasets being upscaled to 90m? I also noticed this discussed in lines 175-176, but there was no justification as to why – please include.

Line 299: "from the multilayer snowpack simulated by MSHM as for the single layer case" *I'm not sure what you are trying to say here*.

Line 366: "restructingthe" – *restricting the*

Figure 7 – None of the panels have a letter denoting which panel they are. The panels with the red box should be made into a new figure that zooms into the distribution of values within the red box – there are discussions in the Results section of what's happening here, but it is difficult for the reader to confirm what the text is saying graphically because it is far too small.

Figure 8, 9, 10, 11 also has no panel labels.

Line 433: "Fig. A7" – where is A7?

Line 445: "Fig A8" – where is A8? Be specific when referring to the appendix.

Line 477: "Fig A11" – where?

Line 534: "including water, forest (4500) and proximity" – *what does 4500 refer to?*

Table 7: *There is a note in the caption that "shaded rows correspond to large local MARE" – however there are no shaded rows in the table.*

**Reply To Reviewer 2**

We thank the Reviewer for the helpful comments and suggestions. The Reviewer's comments are in black. Our replies are in blue.

General Comments

The authors of this manuscript use a Bayesian physical-statistical model to retrieve snow depth using data that was acquired during SnowEx'17, including pit data, ASO lidar acquisitions and the X- and Ku-band SnowSAR data. The methods presented by the authors are clearly defined, with encouraging results with error ranges of about 20cm SWE. The paper is methodologically and analytically sound, but does lack context of why certain methods are being employed, and the representation of results in figures need to be improved. For instance, in the Introduction it was overall well written to describe the need for remote sensing of snow and how SnowEx'17 is an attempt to address this concern.

The Reviewer's comment is well taken. The writing is revised for clarity and detail in Lines 71-81 of the Introduction.

The paper is very detailed in its methods of preparing the data for the model, and the modeling itself. What I think is lacking here is the reason that you are completing some of these steps. For instance, in section 4.1.1. you discuss in detail how you divide the snowpack with multiple layers into a snowpack with 1 or 2 layers, but never state why a 2 layer pack would be useful (that snowpack generally has a wind slab and depth hoar layer). The justification for many of the steps in this paper need to have a bit better context provided.

Thank you. We revised the writing to address these points. In particular, Section 4.1, Lines 250-272 explicitly address the points raised by the Reviewer.

Overall the paper is of publication quality in terms of its research, but the presentation could be improved, with my general and specific comments provided here.

Thank you.

Section 2. Previous Work

Line 55: "Time-series observations are available presently from tower measurements, albeit at the point scale of the tower footprint". I think I know what study/setup you're referring to here, but you have not referenced the papers that have been published based on them. Also, a sentence that states why you are not using a tower approach in this work would be useful.

Line 55 is Line 57 in the revised manuscript: We added a reference, and a sentence for context reading the joint space-time variability of snowpacks.

Line 86: "demonstrated the utility of a couple multi-layer snow hydrology coupled with a.." – do you mean "snow hydrology model coupled..."?

Line 86 is now Line 91: Revised as suggested.

Line 133-134: "the second is the prior of the backscatter… the prior of the snowpack physical…" this sentence is a little unclear, please revise.

Lines 133 -134 are Lines 139-141 in the revised manuscript:  Revised.

Line 144: "assuming that we have good understanding" grammar issue here "assuming that we have a good understanding".

Line 144 is Line 150 in the revised manuscript: Done.

Table 1: I may have missed it, but why are all the datasets being upscaled to 90m? I also noticed this discussed in lines 175-176, but there was no justification as to why – please include.

A justification was added in Section 4.1, lines 259-268 and referred to in Lines 184-185.

Line 299: "from the multilayer snowpack simulated by MSHM as for the single layer case" I'm not sure what you are trying to say here.

Line 299 is Line 336 in the revised manuscript: Removed redundant "as for the single layer case".

Line 366: "restructingthe" – restricting the

Line 366 is Line 396 in the revised manuscript: Revised.

Figure 7 – None of the panels have a letter denoting which panel they are. The panels with the red box should be made into a new figure that zooms into the distribution of values within the red box – there are discussions in the Results section of what's happening here, but it is difficult for the reader to confirm what the text is saying graphically because it is far too small.

Figure 7 was revised to improve readability.  A zoom of the red box region alone is now provided.

Figure 8, 9, 10, 11 also has no panel labels.

Added and reformatted.

Line 433: "Fig. A7" – where is A7?

Line 433 is Line 467 in the revised manuscript: Figure A7 is in Appendix following journal guidelines.  We added an explanation in lines 400-402.

Line 445: "Fig A8" – where is A8 ? Be specific when referring to the appendix.

Line 445 is Line 468 in the revised manuscript: We are following journal guidelines indicating that figures starting by A indicate Appendix.

Line 477: "Fig A11" – where?

See above.

Line 534: "including water, forest (4500) and proximity" – what does 4500 refer to?

Line 534 is Line 517 in the revised manuscript: Revised. 4500 is the reference number of the pit at the bottom of Tables 6 and A2.

Table 7: There is a note in the caption that "shaded rows correspond to large local MARE" – however there are no shaded rows in the table.

Table 7 is Table 6: It should be italicized. Corrected.

Thank you.

[revised manuscript text omitted]

Temperature
Air Pressure
Incoming shortwave radiation
Incoming longwave radiation
Wind speed
Humidity
Albedo | Snow Temperature Profile
Soil Temperature Profile
Snow Density Profile
Snow Depth Layering Profile
Liquid Water Content Profile
Snow Correlation Length Profile | Cao and Barros (2020) |
| MEMLS | Snow Temperature Profile
Soil Temperature Profile
Snow Density Profile
Snow Depth Layering Profile
Snow Correlation Length Profile
Cross polarization fraction
Ground rms height | Diffused Reflectivity Profile
Specular Reflectivity Profile
Total Backscatter Coefficient | Proksch et al. (2015) |
| Base-AM | Equivalent Snow Temperature Prior
Equivalent Soil Temperature Prior
Equivalent Snow Density Prior
Equivalent Snow Depth Prior
Correlation Length
Cross polarization fraction
Ground rms height

[revised manuscript text omitted]